# Off-Policy Evaluation for Action-Dependent Non-Stationary Environments

**Yash Chandak**
University of Massachusetts

**Shiv Shankar**
University of Massachusetts

**Nathaniel D. Bastian**
United States Military Academy

**Bruno Castro da Silva**
University of Massachusetts

**Emma Brunskill**
Stanford University

**Philip S. Thomas**
University of Massachusetts

## Abstract

Methods for sequential decision-making are often built upon a foundational assumption that the underlying decision process is stationary. This limits the application of such methods because real-world problems are often subject to changes due to external factors (*passive* non-stationarity), changes induced by interactions with the system itself (*active* non-stationarity), or both (*hybrid* non-stationarity). In this work, we take the first steps towards the fundamental challenge of on-policy and off-policy evaluation amidst structured changes due to active, passive, or hybrid non-stationarity. Towards this goal, we make a *higher-order stationarity* assumption such that non-stationarity results in changes over time, but the way changes happen is fixed. We propose, OPEN, an algorithm that uses a double application of counterfactual reasoning and a novel importance-weighted instrument-variable regression to obtain both a lower bias and a lower variance estimate of the structure in the changes of a policy's past performances. Finally, we show promising results on how OPEN can be used to predict future performances for several domains inspired by real-world applications that exhibit non-stationarity.

## 1  Introduction

Methods for sequential decision making are often built upon a foundational assumption that the underlying decision process is stationary [Sutton and Barto, 2018]. While this assumption was a cornerstone when laying the theoretical foundations of the field, and while is often reasonable, it is seldom true in practice and can be unreasonable [Dulac-Arnold et al., 2019]. Instead, real-world problems are subject to non-stationarity that can be broadly classified as (a) *Passive:* where the changes to the system are induced only by external (exogenous) factors, (b) *Active:* where the changes result due to the agent's past interactions with the system, and (c) *Hybrid:* where both passive and active changes can occur together [Khetarpal et al., 2020].

There are many applications that are subject to active, passive, or hybrid non-stationarity, and where the stationarity assumption may be unreasonable. Consider methods for automated healthcare where we would like to use the data collected over past decades to find better treatment policies. In such cases, not only might there have been passive changes due to healthcare infrastructure changing over time, but active changes might also occur because of public health continuously evolving based on the treatments made available in the past, thereby resulting in hybrid non-stationarity. Similar to automated healthcare, other applications like online education, product recommendations, and in fact almost all human-computer interaction systems need to not only account for the continually drifting behavior of the user demographic, but also how the preferences of users may change due to interactions with the system [Theocharous et al., 2020]. Even social media platforms need to account for the partisan bias of their users that change due to both external political developments

36th Conference on Neural Information Processing Systems (NeurIPS 2022).

and increased self-validation resulting from previous posts/ads suggested by the recommender system itself [Cinelli et al., 2021, Gillani et al., 2018]. Similarly, motors in a robot suffer wear and tear over time not only based on natural corrosion but also on how vigorous past actions were.

However, conventional off-policy evaluation methods [Precup, 2000, Jiang and Li, 2015, Xie et al., 2019] predominantly focus on the stationary setting. These methods assume availability of either (a) *resetting assumption* to sample multiple sequences of interactions from a stationary environment with a fixed starting state distribution (i.e., episodic setting), or (b) *ergodicity assumption* such that interactions can be sampled from a steady-state/stationary distribution (i.e., continuing setting). For the problems of our interest, methods based on these assumptions may not be viable. For e.g., in automated healthcare, we have a single long history for the evolution of public health, which is neither in a steady state distribution nor can we reset and go back in time to sample another history of interactions.

As discussed earlier, because of non-stationarity the transition dynamics and reward function in the future can be different from the ones in the past, and these changes might also be dependent on past interactions. In such cases, how do we even address the fundamental challenge of *off-policy evaluation*, i.e., using data from past interactions to estimate the performance of a new policy in the future? Unfortunately, if the underlying changes are arbitrary, even amidst only passive non-stationarity it may not be possible to provide non-vacuous predictions of a policy's future performance [Chandak et al., 2020a].

Thankfully, for many real-world applications there might be (unknown) structure in the underlying changes. In such cases, can the *effect* of the underlying changes on a policy's performance be inferred, *without* requiring estimation of the underlying model/process? Prior work has only shown that this is possible in the passive setting. This raises the question that we aim to answer:

> *How can one provide a unified procedure for (off) policy evaluation amidst active,*
> *passive, or hybrid non-stationarity, when the underlying changes are structured?*

**Contributions:** To the best of our knowledge, our work presents the first steps towards addressing the fundamental challenge of off-policy evaluation amidst structured changes due to active or hybrid non-stationarity. Towards this goal, we make a *higher-order stationarity* assumption, under which the non-stationarity can result in changes over time, but the way changes happen is fixed. Under this assumption, we propose a model-free method that can infer the *effect* of the underlying non-stationarity on the past performances and use that to predict the future performances for a given policy. We call the proposed method OPEN: off-policy evaluation for non-stationary domains. On domains inspired by real-world applications, we show that OPEN often provides significantly better results not only in the presence of active and hybrid non-stationarity, but also for the passive setting where it even outperforms previous methods designed to handle only passive non-stationarity.

OPEN primarily relies upon two key insights: **(a)** For active/hybrid non-stationarity, as the underlying changes may dependend on past interactions, the structure in the changes observed when executing the data collection policy can be different than if one were to execute the evaluation policy. To address this challenge, OPEN makes uses counterfactual reasoning twice and permits reduction of this off-policy evaluation problem to an auto-regression based forecasting problem. **(b)** Despite reduction to a more familiar auto-regression problem, in this setting naive least-squares based estimates of parameters for auto-regression suffers from high variance and can even be asymptotically biased. Finally, to address this challenge, OPEN uses a novel importance-weighted instrument-variable (auto-)regression technique to obtain asymptotically consistent and lower variance parameter estimates.

## 2   Related Work

Off-policy evaluation (OPE) is an important aspect of reinforcement learning [Precup, 2000, Thomas et al., 2015, Sutton and Barto, 2018] and various techniques have been developed to construct efficient estimators for OPE [Jiang and Li, 2015, Thomas and Brunskill, 2016, Munos et al., 2016, Harutyunyan et al., 2016, Espeholt et al., 2018, Xie et al., 2019]. However, these work focus on the stationary setting. Similarly, there are various methods for tackling non-stationarity in the bandit setting [Moulines, 2008, Besbes et al., 2014, Seznec et al., 2018, Wang et al., 2019a]. In contrast, the proposed work focuses on methods for sequential decision making.

Literature on off-policy evaluation amidst non-stationarity for sequential decision making is sparse. Perhaps the most closely related works are by Thomas et al. [2017], Chandak et al. [2020b], Xie et al. [2020a], Poiani et al. [2021], Liotet et al. [2021]. While these methods present an important stepping stone, such methods are for passive non-stationarity and, as we discuss using the toy example in Figure 1, may result in undesired outcomes if used as-is in real-world settings that are subject to active or hybrid non-stationarity.

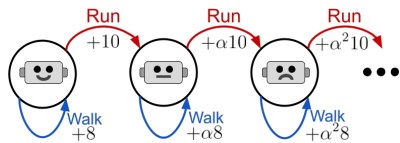

Figure 1: RoboToy domain.

Consider a robot that can perform a task each day either by 'walking' or 'running'. A reward of 8 is obtained upon completion using 'walking', but 'running' finishes the task quickly and results in a reward of 10. However, 'running' wears out the motors, thereby increasing the time to finish the task the next day and reduces the returns for *both* 'walking' and 'running' by a small factor, $\alpha \in (0, 1)$.

Here, methods for tackling *passive* non-stationarity will track the best policy under the assumption that the changes due to damages are because of external factors and would fail to attribute the cause of damage to the agent's decisions. Therefore, as on any given day 'running' will always be better, every day these methods will prefer 'running' over 'walking' and thus *aggravate* the damage. Since the outcome on each day is dependent on decisions made during previous days this leads to active non-stationarity, where 'walking' is better in the long run. Finding a better policy first requires a method to evaluate a policy's (future) performance, which is the focus of this work.

Notice that the above problem can also be viewed as a task with effectively a *single* lifelong episode. However, as we discuss later in Section 4, approaches such as modeling the problem as a large stationary POMDP or as a continuing average-reward MDP with a single episode may not be viable. Further, non-stationarity can also be observed in multi-agent systems and games due to different agents/players interacting with the system. However, often the goal in these other areas is to search for (Nash) equilibria, which may not even exist under hybrid non-stationarity. Non-stationarity may also result due to artifacts of the learning algorithm even when the problem is stationary. While relevant, these other research areas are distinct from our setting of interest and we discuss them and others in more detail in Appendix B.

## 3 Non-Stationary Decision Processes

We build upon the formulation used by past work [Xie et al., 2020a, Chandak et al., 2020b] and consider that the agent interacts with a lifelong sequence of partially observable Markov decision processes (POMDPs), $(M_i)_{i=1}^{\infty}$. However, unlike prior problem formulations, we account for active and hybrid non-stationarity by considering POMDP $M_{i+1}$ to be dependent on *both* on the POMDP $M_i$ and the decisions made by the agent while interacting with $M_i$. We provide a control graph for this setup in Figure 2. For simplicity of presentation, we will often ignore the dependency of $M_{i+1}$ on $M_{i-k}$ for $k > 0$, although our results can be extended for settings with $k > 0$.

**Notation:** Let $\mathcal{M}$ be a finite set of POMDPs. Each POMDP $M_i \in \mathcal{M}$ is a tuple $(\mathcal{O}, \mathcal{S}, \mathcal{A}, \Omega_i, P_i, R_i, \mu_i)$, where $\mathcal{O}$ is the set of observations, $\mathcal{S}$ is the set of states, and $\mathcal{A}$ is the set of actions, which are the same for all the POMDPs in $\mathcal{M}$. For simplicity of notation, we assume $\mathcal{M}, \mathcal{S}, \mathcal{O}, \mathcal{A}$ are finite sets, although our results can be extended to settings where these sets are infinite or continuous. Let $\Omega_i : \mathcal{S} \times \mathcal{O} \to [0, 1]$ be the observation function, $P_i : \mathcal{S} \times \mathcal{A} \times \mathcal{S} \to [0, 1]$ be the transition function, $\mu_i : \mathcal{S} \to [0, 1]$ be the starting state distribution, and $R_i : \mathcal{S} \times \mathcal{A} \to [-R_{\max}, R_{\max}]$ be the reward function with $0 \leq R_{\max} < \infty$.

Let $\pi : \mathcal{O} \times \mathcal{A} \to [0, 1]$ be any policy and $\Pi$ be the set of all policies. Let $H_i := (O_i^t, A_i^t, R_i^t)_{t=1}^T$ be a sequence of at most $T$ interactions in $M_i$, where $O_i^t, A_i^t, R_i^t$ are the random variables corresponding to the observation, action, and reward at the step $t$. Let $G_i := \sum_{t=1}^T R_i^t$ be an observed return and $J_i(\pi) := \mathbb{E}_\pi[G_i|M_i]$ be the performance of $\pi$ on $M_i$. Let $\mathcal{H}$ be the set of possible interaction sequences, and finally let $\mathcal{T} : \mathcal{M} \times \mathcal{H} \times \mathcal{M} \to [0, 1]$ be the transition function that governs the non-stationarity in the POMDPs. That is, $\mathcal{T}(m, h, m') = \Pr(M_{i+1}{=}m'|M_i{=}m, H_i{=}h)$.

Figure 2 (**Left**) depicts the control graph for a stationary POMDP, where each column corresponds to one time step. Here, *multiple, independent* episodes from the *same* POMDP can be resampled. (**Right**) Control graph that we consider for a non-stationary decision process, where each column corresponds to one episode. Here, the agent interacts with a *single* sequence of related POMDPs $(M_i)_{i=1}^n$. Absence or presence of the red arrows indicates whether the change from $M_i$ to $M_{i+1}$ is independent of the decisions in $M_i$ (passive non-stationarity) or not (active non-stationarity).

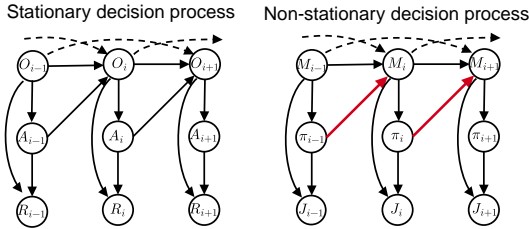

Figure 2: Control Graph for the non-stationary process. See text for symbol definitions.

**Problem Statement:** We look at the fundamental problem of evaluating the performance of a policy $\pi$ in the presence of non-stationarity. Let $(H_i)_{i=1}^n$ be the data collected in the past by interacting using policies $(\beta_i)_{i=1}^n$. Let $D_n$ be the dataset consisting of $(H_i)_{i=1}^n$ and the probabilities of the actions taken by $(\beta_i)_{i=1}^n$. Given $D_n$, we aim to evaluate the expected future performance of $\pi$ if it is deployed for the *next* $L$ episodes (each a different POMDP), that is $\mathscr{J}(\pi) \coloneqq \mathbb{E}_\pi \left[ \sum_{k=n+1}^{n+L} J_k(\pi) \Big| (H_i)_{i=1}^n \right]$. We call it the *on-policy* setting if $\forall i, \beta_i = \pi$, and the *off-policy* setting otherwise. Notice that even in the on-policy setting, naively aggregating observed performances from $(H_i)_{i=1}^n$ may not be indicative of $\mathscr{J}(\pi)$ as $M_k$ for $k > n$ may be different than $M \in (M_i)_{i=1}^n$ due to non-stationarity.

## 4 Understanding Structural Assumptions

A careful reader would have observed that instead of considering interactions with a sequence of POMDPs $(M_i)_{i=1}^n$ that are each dependent on the past POMDPs and decisions, an equivalent setup might have been to consider a 'chained' sequence of interactions $(H_1, H_2, ..., H_n)$ as a *single* episode in a 'mega' POMDP comprised of all $M \in \mathcal{M}$. Consequently, $\mathscr{J}(\pi)$ would correspond to the expected future return given $(H_i)_{i=1}^n$. Tackling this single long sequence of interactions using the continuing/average-reward setting is not generally viable because methods for these settings rely on an ergodicity assumption (which implies that all states can always be revisited) that may not hold in the presence of non-stationarity. For instance, in the earlier example of automated healthcare, it is not possible to revisit past years.

To address the above challenge, we propose introducing a different structural assumption. Particularly, framing the problem as a sequence of POMDPs allows us to split the single sequence of interactions into multiple (dependent) fragments, with additional structure linking together the fragments. Specifically, we make the following intuitive assumption.

**Assumption 1.** $\forall m \in \mathcal{M}$ *such that the performance $J(\pi)$ associated with $m$ is $j$,*

$$\forall \pi, \pi' \in \Pi^2, \forall i, \ \Pr(J_{i+1}(\pi) = j_{i+1} | M_i = m; \pi') = \Pr(J_{i+1}(\pi) = j_{i+1} | J_i(\pi) = j; \pi'). \quad (1)$$

Assumption 1 characterizes the probability that $\pi$'s performance will be $j_{i+1}$ in the $i+1^{\text{th}}$ episode when the policy $\pi'$ is executed in the $i^{\text{th}}$ episode. To understand Assumption 1 intuitively, consider a 'meta-transition' function that characterizes $\Pr(J_{i+1}(\pi) | J_i(\pi), \pi')$ similar to how the standard transition function in an MDP characterizes $\Pr(S_{t+1} | S_t, A_t)$. While the underlying changes actually happen via $\mathcal{T}$, Assumption 1 imposes the following two conditions: **(a)** A *higher-order stationarity* condition on the meta-transitions under which non-stationarity can result in changes over time, but *the way the changes happen is fixed*, and **(b)** Knowing the past performance(s) of a policy $\pi$ provides *sufficient* information for the meta-transition function to model how the performance will change upon executing any (possibly different) policy $\pi'$. For example, in the earlier toy robot domain, given the current performance there exists an (unknown) oracle that can predict the performance for the next day if the robot decides to 'run'/'walk'.

Assumption 1 is beneficial as it implicitly captures the effect of both the underlying passive and active non-stationarity by modeling the conditional distribution of the performance $J_{i+1}(\pi)$ given $J_i(\pi)$, when executing any (different) policy $\pi'$. At the same time, notice that it generalizes **(a)** the stationary setting, where $\forall \pi \in \Pi, \forall i > 0, J_{i+1}(\pi) = J_i(\pi)$, and **(b)** only passive non-stationarity,

which is a special case of (1) wherein $\pi'$ does not influence the outcome, i.e.,

$$\forall \pi_a, \pi_b \in \Pi^2, \forall i > 0, \Pr(J_{i+1}(\pi) = J_{i+1}|J_i(\pi) = j; \textcolor{red}{\pi_a}) = \Pr(J_{i+1}(\pi) = J_{i+1}|J_i(\pi) = j; \textcolor{red}{\pi_b}).$$

**Remark 1.** *In some cases, it may be beneficial to relax Assumption 1 such that instead of using* $\Pr(J_{i+1}(\pi)|J_i(\pi); \pi')$ *in (1), one considers* $\Pr(J_{i+1}(\pi)|(J_{i-k}(\pi))_{k=0}^p; \pi')$. *This can be considered similar to the p-Markov MDP where the transitions are characterized using* $\Pr(S_{t+1}|(S_{t-i})_{i=0}^p, A_t)$. *While we consider this general setting for our empirical results, for simplicity, to present the key ideas we will consider (1). We provide a detailed discussion on cases where we expect such an assumption to be* (in)valid, *and also other potential assumptions in Appendix C.*

## 5 Model-Free Policy Evaluation

In this section we discuss how under Assumption 1, we can perform model-free off-policy evaluation amidst passive, active, or hybrid non-stationarity. The high level idea can be decomposed into the following: **(a)** Obtain estimates of $(J_i(\pi))_{i=1}^n$ using $(H_i)_{i=1}^n$ (red arrows in Figure 3), and **(b)** Use the estimates of $(J_i(\pi))_{i=1}^n$ to infer the *effect* of the underlying non-stationarity on the performance, and use that to predict $(J_i(\pi))_{i=n+1}^{n+L}$ (blue arrows in Figure 3).

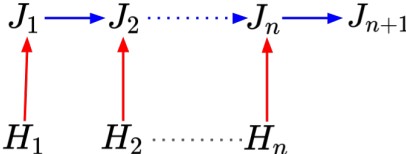

Figure 3: High-level idea.

**5.1 Counterfactual Reasoning** $(J_i(\pi))_{i=1}^n$ could have been directly estimated if we had access to $(M_i)_{i=1}^n$. However, how do we estimate $(J_i(\pi))_{i=1}^n$ when we only have $(H_i)_{i=1}^n$ collected using interactions via possibly different data collecting policies $(\beta_i)_{i=1}^n$?

To estimate $(J_i(\pi))_{i=1}^n$, we use the collected data $D_n$ and aim to answer the following counterfactual question: *what would the performance of $\pi$ would have been, if $\pi$ was used to interact with $M_i$ instead of $\beta_i$?* To answer this, we make the following standard support assumption [Thomas et al., 2015, Thomas and Brunskill, 2016, Xie et al., 2019] that says that any action that is likely under $\pi$ is also sufficiently likely under the policy $\beta_i$ for all $i$.

**Assumption 2.** $\forall o \in \mathcal{O}, \forall a \in \mathcal{A}$, and $\forall i \leq n$, $\frac{\pi(o,a)}{\beta_i(o,a)}$ is bounded above by a (unknown) constant c.

Under Assumption 2, an unbiased estimate of $J_i(\pi)$ can be obtained using common off-policy evaluation methods like importance sampling (IS) or per-decision importance sampling (PDIS) [Precup, 2000], $\forall i, \widehat{J}_i(\pi) := \sum_{t=1}^T \rho_i^t R_i^t$, where, $\rho_i^t := \prod_{j=1}^t \frac{\pi(O_i^j, A_i^j)}{\beta_i(O_i^j, A_i^j)}$. This $\widehat{J}_i(\pi)$ provides an estimate of $J_i(\pi)$ associated with each $M_i$ and policy $\pi$, as needed for the red arrows in Figure 3.

**5.2 Double Counterfactual Reasoning** Having obtained the estimates for $(J_i(\pi))_{i=1}^n$, we now aim to estimate how the performance of $\pi$ changes due to the underlying non-stationarity. Recall that under active or hybrid non-stationarity, changes in a policy's performance due to the underlying non-stationarity is dependent on the past actions. From Assumption 1, let

$$\forall i > 0, \quad F_\pi(x, \pi', y) := \Pr(J_{i+1}(\pi) = y|J_i(\pi) = x; \pi')$$

denote how the performance of $\pi$ changes between episodes, if $\pi'$ was executed. Here $J_{i+1}(\pi)$ is a random variable because of stochasticity in $H_i$ (i.e., how $\pi'$ interacts in $M_i$), as well as in the meta-transition from POMDP $M_i$ to $M_{i+1}$. Similarly, let

$$\forall i > 0, \quad f(J_i(\pi), \pi'; \theta_\pi) := \mathbb{E}_{\pi'}[J_{i+1}(\pi)|J_i(\pi)] = \sum_{y \in \mathbb{R}} F_\pi(J_i(\pi), \pi', y)y$$

be some (unknown) function parameterized by $\theta_\pi \in \Theta$, which denotes the *expected* performance of $\pi$ in episode $i+1$, if in episode $i$, $\pi$'s performance was $J_i(\pi)$ and $\pi'$ was executed. Parameters $\theta_\pi$ depend on $\pi$ and thus $f$ can model different types of changes to the performance of different policies.

Recall from Figure 3 (blue arrows), if we can estimate $f(\cdot, \pi; \theta_\pi)$ to infer how $J_i(\pi)$ changes due to the underlying non-stationarity when interacting with $\pi$, then we can use it to predict $(J_i(\pi))_{i=n+1}^{n+L}$ when $\pi$ is deployed in the future. In the following, we will predominantly focus on estimating $f(\cdot, \pi; \theta_\pi)$ using past data $D_n$. Therefore, for brevity we let $f(\cdot; \theta_\pi) := f(\cdot, \pi; \theta_\pi)$.

If pairs of $(J_i(\pi), J_{i+1}(\pi))$ are available when the transition between $M_i$ and $M_{i+1}$ occurs due to execution of $\pi$, then one could auto-regress $J_{i+1}(\pi)$ on $J_i(\pi)$ to estimate $f(\cdot; \theta_\pi)$ and model the changes in the performance of $\pi$. However, the sequence $(\widehat{J}_i(\pi))_{i=1}^n$ obtained from counterfactual reasoning cannot be used as-is for auto-regression. This is because the changes that occurred between $M_i$ and $M_{i+1}$ are associated with the execution of $\beta_i$, not $\pi$. For example, recall the toy robot example in Figure 1. If data was collected by mostly 'running', then the performance of 'walking' would decay as well. Directly auto-regressing on the past performances of 'walking' would result in how the performance of 'walking' would change *when actually executing 'running'*. However, if we want to predict performances of 'walking' in the future, what we actually want to estimate is how the performance of 'walking' changes *if 'walking' is actually performed*.

To resolve the above issue, we ask another counter-factual question: *What would the performance of $\pi$ in $M_{i+1}$ have been had we executed $\pi$, instead of $\beta_i$, in $M_i$?* In the following theorem we show how this question can be answered with a second application of the importance ratio $\rho_i := \rho_i^T$.

**Theorem 1.** *Under Assumptions* 1 *and* 2, $\forall m \in \mathcal{M}$ *such that the performance $J(\pi)$ associated with $m$ is $j$,* $\mathbb{E}_\pi\left[J_{i+1}(\pi)|J_i(\pi) = j\right] = \mathbb{E}_{\beta_i, \beta_{i+1}}\left[\rho_i \widehat{J}_{i+1}(\pi)\big|M_i = m\right].$

See Appendix D.1 for the proof. Intuitively, as $\beta_i$ and $\beta_{i+1}$ were used to collect the data in $i$ and $i + 1^{\text{th}}$ episodes, respectively, Theorem 1 uses $\rho_i$ to first correct for the mismatch between $\pi$ and $\beta_i$ that influences how $M_i$ changes to $M_{i+1}$ due to interactions $H_i$. Secondly, $\widehat{J}_{i+1}$ corrects for the mismatch between $\pi$ and $\beta_{i+1}$ for the sequence of interactions $H_{i+1}$ in $M_{i+1}$.

**5.3 Importance-Weighted IV-Regression**    An important advantage of Theorem 1 is that given $J_i(\pi)$, $\rho_i \widehat{J}_{i+1}(\pi)$ provides an unbiased estimate of $\mathbb{E}_\pi\left[J_{i+1}(\pi)|J_i(\pi)\right]$, even though $\pi$ may not have been used for data collection. This permits using $Y_i := \rho_i \widehat{J}_{i+1}(\pi)$ as a target for predicting the next performance given $X_i := J_i(\pi)$, i.e., to estimate $f(J_i(\pi); \theta_\pi)$ through regression on $(X_i, Y_i)$ pairs.

However, notice that performing regression on the pairs $(X_i = J_i(\pi), Y_i = \rho \widehat{J}_{i+1}(\pi))_{i=1}^{n-1}$ may not be directly possible as we do not have $J_i(\pi)$; only unbiased *estimates* $\widehat{J}_i(\pi)$ of $J_i(\pi)$. This is problematic because in least-squares regression, while noisy estimates of the *target* variable $Y_i$ are fine, noisy estimates of the *input* variable $X_i$ may result in estimates of $\theta_\pi$ that are *not even* asymptotically consistent *even* when the underlying $f$ is a linear function of its inputs. To see this clearly, consider the following naive estimator,

$$\hat{\theta}_{\texttt{naive}} \in \underset{\theta \in \Theta}{\arg\min} \ \sum_{i=1}^{n-1}\left(f\left(\widehat{J}_i(\pi); \theta\right) - \rho_i \widehat{J}_{i+1}(\pi)\right)^2.$$

Because $\widehat{J}_i(\pi)$ is an unbiased estimate of $J_\pi$, without loss of generality, let $\widehat{J}_i(\pi) = J_i(\pi) + \eta_i$, where $\eta_i$ is mean zero noise. Let $\mathbb{N} := [\eta_1, \eta_2, ..., \eta_{n-1}]^\top$ and $\mathbb{J} := [J_1(\pi), J_2(\pi), ..., J_{n-1}(\pi)]^\top$. $\theta_{\texttt{naive}}$ can now be expressed as (see Appendix D.2),

$$\hat{\theta}_{\texttt{naive}} \xrightarrow{a.s.} \left(\mathbb{J}^\top \mathbb{J} + \mathbb{N}^\top \mathbb{N}\right)^{-1} \mathbb{J}^\top \mathbb{J} \theta_\pi \ \xcancel{\xrightarrow{a.s.}} \ \theta_\pi. \tag{2}$$

Observe that $\mathbb{N}^\top \mathbb{N}$ in (2) relates to the variances of the mean zero noise variables $\eta_i$. The greater the variances, the more $\hat{\theta}_{\texttt{naive}}$ would be biased towards zero (if $\forall i$, $\eta_i = 0$, then the true $\theta_\pi$ is trivially recovered). Intuitively, when the variance of $\eta_i$ is high, noise dominates and the structure in the data gets suppressed even in the large-sample regime. Unfortunately, the importance sampling based estimator $\widehat{J}_i(\pi)$ in the sequential decision making setting is infamous for extremely high variance [Thomas et al., 2015]. Therefore, $\hat{\theta}_{\texttt{naive}}$ can be extremely biased and will not be able to capture the trend in how performances are changing, *even in the limit of infinite data and linear $f$*. The problem may be exacerbated when $f$ is non-linear.

**5.3.1 Bias Reduction**    To mitigate the bias stemming from noise in input variables, we introduce a novel instrument variable (IV) [Pearl et al., 2000] regression method for tackling non-stationarity. Instrument variables $Z$ represent some side-information and were originally used in the causal literature to mitigate any bias resulting due to spurious correlation, caused by unobserved confounders, between the input and the target variables. For mitigating bias in our setting, IVs can intuitively be considered as some side-information to 'denoise' the input variable before performing regression.

For this IV-regression, an ideal IV is *correlated* with the input variables (e.g., $\widehat{J}_i(\pi)$) but *uncorrelated* with the noises in the input variable (e.g., $\eta_i$).

We propose leveraging statistics based on past performances as an IV for $\widehat{J}_i(\pi)$. For instance, using $Z_i \coloneqq \widehat{J}_{i-1}(\pi)$ as an IV for $\widehat{J}_i(\pi)$. Notice that while correlation between $J_{i-1}(\pi)$ and $J_i(\pi)$ can directly imply correlation between $\widehat{J}_{i-1}(\pi)$ and $\widehat{J}_i(\pi)$, values of $J_{i-1}(\pi)$ and $J_i(\pi)$ are dependent on non-stationarity in the past. Therefore, we make the following assumption, which may easily be satisfied when the consecutive performances do not change arbitrarily.

**Assumption 3.** $\forall i, \quad \mathrm{Cov}\left(\widehat{J}_{i-1}(\pi), \widehat{J}_i(\pi)\right) \neq 0.$

However, notice that the noise in $\widehat{J}_i(\pi)$ can be *dependent* on $\widehat{J}_{i-1}(\pi)$. This is because non-stationarity can make $H_{i-1}$ and $H_i$ dependent, which are in turn used to estimate $\widehat{J}_{i-1}(\pi)$ and $\widehat{J}_i(\pi)$, respectively. Nevertheless, perhaps interestingly, we show that despite not being independent, $\widehat{J}_{i-1}(\pi)$ is *uncorrelated* with the noise in $\widehat{J}_i(\pi)$.

**Theorem 2.** *Under Assumptions* 1 *and* 2, $\forall i, \quad \mathrm{Cov}\left(\widehat{J}_{i-1}(\pi), \widehat{J}_i(\pi) - J_i(\pi)\right) = 0.$

See Appendix D.3 for the proof. Finally, as IV regression requires learning an additional function $g \coloneqq \mathbb{R} \to \mathbb{R}$ parameterized by $\varphi \in \Omega$ (intuitively, think of this as a denoising function), we let $\widehat{J}_{i-1}(\pi)$ be an IV for $\widehat{J}_i(\pi)$ and propose the following IV-regression based estimator,

$$\hat{\varphi}_n \in \operatorname*{argmin}_{\varphi \in \Omega} \sum\nolimits_{i=2}^n \left(g\left(\widehat{J}_{i-1}(\pi); \varphi\right) - \widehat{J}_i(\pi)\right)^2 \tag{3}$$

$$\hat{\theta}_n \in \operatorname*{argmin}_{\theta \in \Theta} \sum\nolimits_{i=2}^{n-1} \left(f\left(g\left(\widehat{J}_{i-1}(\pi); \hat{\varphi}_n\right); \theta\right) - \rho_i \widehat{J}_{i+1}(\pi)\right)^2. \tag{4}$$

**Theorem 3.** *Under Assumptions* 1, 2, *and* 3, *if $f$ and $g$ are linear functions of their inputs, then $\hat{\theta}_n$ is a strongly consistent estimator of $\theta_\pi$, i.e., $\hat{\theta}_n \xrightarrow{a.s.} \theta_\pi$. (See Appendix D.3 for the proof.)*

**Remark 2.** *Other choices of instrument variables $Z_i$ (apart from $Z_i = \widehat{J}_{i-1}(\pi)$) are also viable. We discuss some alternate choices in Appendix E. These other IVs can be used in (3) and (4) by replacing $\widehat{J}_{i-1}(\pi)$ with the alternative $Z_i$.*

**Remark 3.** *As discussed earlier, it may be beneficial to model $J_{i+1}(\pi)$ using $(J_k(\pi))_{k=i-p+1}^i$ with $p > 1$. The proposed estimator can be easily extended by making $f$ dependent on multiple past terms $(X_k)_{k=i-p+1}^i$, where $\forall k, X_k \coloneqq g((\widehat{J}_l(\pi))_{l=k-p}^{k-1}; \hat{\phi})$. We discuss this in more detail in Appendix E. The proposed procedure is also related to methods that use lags of the time series as instrument variables [Bellemare et al., 2017, Wilkins, 2018, Wang and Bellemare, 2019].*

**Remark 4.** *An advantage of the model-free setting is that we only need to consider changes in $J(\pi)$, which is a **scalar** statistic. For scalar quantities,* linear *auto-regressive models have been known to be useful in modeling a wide variety of time-series trends. Nonetheless,* non-linear *functions like RNNs and LSTMs [Hochreiter and Schmidhuber, 1997] may also be leveraged using deep instrument variable methods [Hartford et al., 2017, Bennett et al., 2019, Liu et al., 2020, Xu et al., 2020].*

As required for the blue arrows in Figure 3, $f(\cdot; \hat{\theta}_n)$ can now be used to estimate the expected value $\mathbb{E}_\pi[J_{i+1}(\pi)|J_i(\pi)]$ under hybrid non-stationarity. Therefore, using $f(\cdot; \hat{\theta}_n)$ we can now auto-regressively forecast the future values of $(J_i(\pi))_{i=n+1}^{n+L}$ and obtain an estimate for $\mathscr{J}(\pi)$. A complete algorithm for the proposed procedure is provided in Appendix E.1.

**5.3.2 Variance Reduction** As discussed earlier, importance sampling results in noisy estimates of $J_i(\pi)$. During regression, while high noise in the input variable leads to high bias, high noise in the target variables leads to high variance parameter estimates. Unfortunately, (3) and (4) have target variables containing $\rho_i$ (and $\rho_{i+1}$) which depend on the product of importance ratios and can thus result in extremely large values leading to higher variance parameter estimates.

The instrument variable technique helped in mitigating bias. To mitigate variance, we draw inspiration from the reformulation of weighted-importance sampling presented for the *stationary* setting by

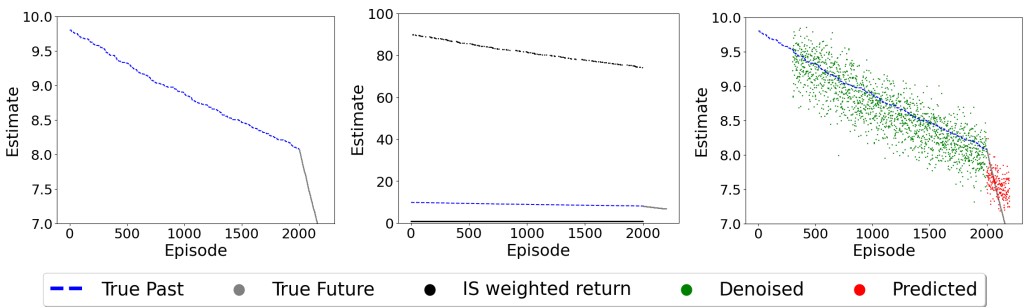

Figure 4: An illustration of the stages in the proposed method for the RoboToy domain of Figure 1. Here, evaluation policy $\pi$ chooses to 'run' more often, whereas the data collecting policy $\beta$ chooses to 'walk' more often. **(Left)** This results in a slow decline of performance for $\pi$ initially, followed by a faster decline once $\pi$ is deployed after episode 2000. The blue and gray curves are unknown to the algorithm. **(Middle)** OPEN first uses historical data to obtain counterfactual estimates of $J_i(\pi)$ for the past episodes. One can see the high-variance in these estimates **(notice the change in the y-scale)** due to the use of importance sampling. **(Right)** Intuitively, before naively auto-regressing, OPEN first denoises past performance estimates using the first stage of IV regression (i.e., converts black dots to green dots). It can be observed that OPEN successfully denoises the importance sampling estimates. Using these denoised estimates and a second use of counterfactual reasoning, OPEN performs the second stage of IV regression. It is able to estimate that once $\pi$ is deployed, performances in the future will decrease more rapidly compared to what was observed in the past.

Mahmood et al. [2014], and propose the following estimator,

$$\tilde{\varphi}_n \in \operatorname*{argmin}_{\varphi \in \Omega} \sum_{i=2}^{n} \bar{\rho}_i \left( g\left( \widehat{J}_{i-1}(\pi); \varphi \right) - G_i(\pi) \right)^2, \qquad \text{where} \quad \bar{\rho}_i := \frac{\rho_i}{\sum_{j=2}^{n} \rho_j} \quad (5)$$

$$\tilde{\theta}_n \in \operatorname*{argmin}_{\theta \in \Theta} \sum_{i=2}^{n-1} \rho_i^\dagger \left( f\left( g\left( \widehat{J}_{i-1}(\pi); \tilde{\varphi}_n \right); \theta \right) - G_{i+1}(\pi) \right)^2, \quad \text{where} \quad \rho_i^\dagger := \frac{\rho_i \rho_{i+1}}{\sum_{j=2}^{n-1} \rho_j \rho_{j+1}} \quad (6)$$

where $G_i$ is the return observed for $M_i$. Intuitively, instead of importance weighting the *target*, we importance weight the squared error, proportional to how likely that *error* would be if $\pi$ was used to collect the data. Since dividing by any constant does not affect $\tilde{\varphi}_n$ and $\tilde{\theta}_n$, the choice of $\bar{\rho}_i$ and $\rho_i^\dagger$ ensures that both $\bar{\rho}_i$ and $\rho_i^\dagger \in [0, 1]$, thereby mitigating variance but still providing consistency.

**Theorem 4.** *Under Assumptions 1, 2, and 3, if $f$ and $g$ are linear functions of their inputs, then $\tilde{\theta}_n$ is a strongly consistent estimator of $\theta_\pi$, i.e., $\tilde{\theta}_n \xrightarrow{a.s.} \theta_\pi$. (See Appendix D.3 for the proof.)*

## 6 Empirical Analysis

This section presents both qualitative and quantitative empirical evaluations using several environments inspired by real-world applications that exhibit non-stationarity. In the following paragraphs, we first briefly discuss different algorithms being compared and answer three primary questions.[1]

**1. OPEN:** We call our proposed method OPEN: off-policy evaluation for non-stationary domains with structured passive, active, or hybrid changes. It is based on our bias and variance reduced estimator developed in (5) and (6). Appendix E.1 contains the complete algorithm.

**2. Pro-WLS:** For the baseline, we use Prognosticator with weighted least-squares (Pro-WLS) [Chandak et al., 2020b]. This method is designed to tackle only passive non-stationarity.

**3. WIS:** A weighted importance sampling based estimator that ignores presence of non-stationarity completely [Precup, 2000].

**4. SWIS:** Sliding window extension of WIS which instead of considering all the data, only considers data from the recent past.

---

[1]Code is available at https://github.com/yashchandak/activeNS

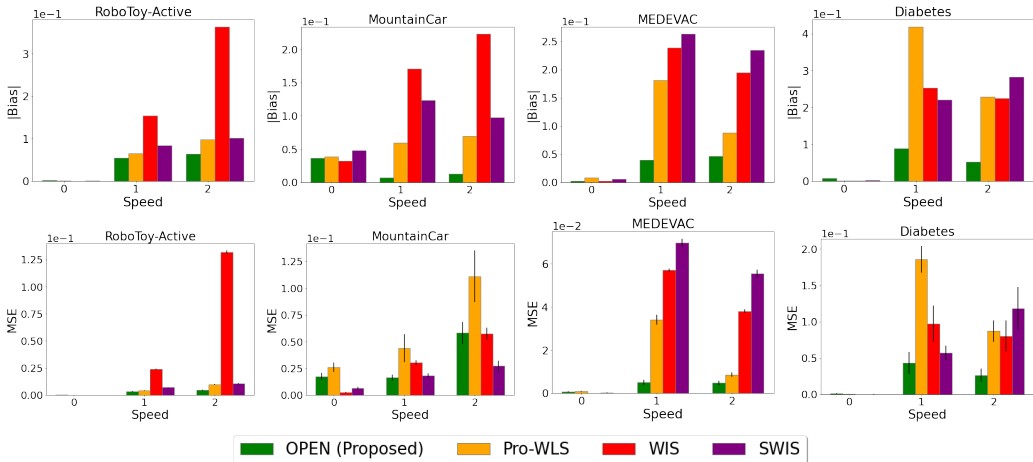

Figure 5: Comparison of different algorithms for predicting the future performance of evaluation policy $\pi$ on domains that exhibit active/hybrid non-stationarity. On the x-axis is the speed which corresponds to the rate of non-stationarity; higher speed indicates faster rate of change and a speed of zero indicates stationary domain. On the y-axes are the absolute bias **(Top row)** and the mean-squared error **(Bottom row)** of the predicted performance estimate *(lower is better everywhere)*. For each domain, for each speed, for each algorithm, 30 trials were executed.

### *Q1. (Qualitative Results) What is the impact of the two stages of the OPEN algorithm?*

In Figure 4 we present a step by step breakdown of the intermediate stages of a single run of OPEN on the RoboToy domain from Figure 1. It can be observed that OPEN is able to extract the effect of the underlying active non-stationarity on the performances and also detect that the evaluation policy $\pi$ that 'runs' more often will cause an active harm, if deployed in the future.

### *Q2. (Quantitative Results) What is the effect of different types and rates of non-stationarity?*

Besides the toy robot from Figure 1, we provide empirical results on three other domains inspired by real-world applications that exhibit non-stationarity. Appendix E.3 contains details for each, including how the evaluation policy and the data collecting policy were designed for them.

**Non-stationary Mountain Car:** In real-world mechanical systems, motors undergo wear and tear over time based on how vigorously they have been used in the past. To simulate similar performance degradation, we adapt the classic (stationary) mountain car domain [Sutton and Barto, 2018]. We modify the domain such that after every episode the effective acceleration force is decayed proportional to the average velocity of the car in the current episode. This results in active non-stationarity, where the change in the system is based on the actions taken by the agent in the past.

**Type-1 Diabetes Management:** Personalised automated healthcare systems for individual patients should account for the physiological and lifestyle changes of the patient over time. To simulate such a scenario we use an open-source implementation [Xie, 2019] of the U.S. Food and Drug Administration (FDA) approved Type-1 Diabetes Mellitus simulator (T1DMS) [Man et al., 2014] for the treatment of Type-1 diabetes, where we induced non-stationarity by oscillating the body parameters (e.g., rate of glucose absorption, insulin sensitivity, etc.) between two known configurations available in the simulator. This induces passive non-stationarity, that is, changes are not dependent on past actions.

**MEDEVAC:** This domain stands for medical evacuation using air ambulances. This domain was developed by Robbins et al. [2020] for optimally routing air ambulances to provide medical assistance in regions of conflict. Based on real-data, this domain simulates the arrival of different events, from different zones, where each event can have different priority levels. Serving higher priority events yields higher rewards. A good controller decides whether to deploy, and which MEDEVAC to deploy, to serve any event (at the risk of not being able to serve a new high-priority event if all ambulances become occupied). Here, the arrival rates of different events can change based on external incidents during conflict. Similarly, the service completion rate can also change based on how frequently an ambulance is deployed in the past. To simulate such non-stationarity, we oscillate the arrival rate of

the incoming high-priority events, which induces passive non-stationarity. Further, to induce wear and tear, we decay the service rate of an ambulance proportional to how frequently the ambulance was used in the past. This induces active non-stationarity. The presence of both active and passive changes makes this domain subject to hybrid non-stationarity.

Figure 5 presents the (absolute) bias and MSE incurred by different algorithms for predicting the future performance of the evaluation policy $\pi$. As expected, the baseline method WIS that ignores the non-stationarity completely fails to capture the change in performances over time. Therefore, while WIS works well for the stationary setting, as the rate of non-stationarity increases, the bias incurred by WIS grows. In comparison, the baseline method Pro-WLS that can only account for passive non-stationarity captures the trend better than WIS, but still performs poorly in comparison to the proposed method OPEN that is explicitly designed to handle active/hybrid non-stationarity. Perhaps interestingly, for the Diabetes domain which only has passive non-stationarity, we observe that OPEN performs better than Pro-WLS. As we discuss later, this can be attributed to the sensitivity of Pro-WLS to its hyper-parameters.

While OPEN incorporated one variance reduction technique, it can be noticed when the rate of non-stationarity is high, variance can sometimes still be high thereby leading to higher MSE. We discuss potential negative impacts of this in Appendix A. Incorporating (partial) knowledge of the underlying model and developing doubly-robust version of OPEN could potentially mitigate variance further. We leave this extension for future work.

### Q3. (Ablations Results) How robust are the methods to hyper-parameters?

Due to space constraints, we defer the empirical results and discussion for this to Appendix E.5. Overall, we observe that the proposed method OPEN being an auto-regressive method can extrapolate/forecast better and is thus more robust to hyper-parameters (number of past terms to condition, as discussed in Remark 3) than Pro-WLS that uses Fourier bases for regression (where the hyper-parameter is the order of Fourier basis) and is not as good for extrapolation.

## 7    Conclusion

We took the first steps for addressing the fundamental question of off-policy evaluation under the presence of non-stationarity. Towards this goal we discussed the need for structural assumptions and developed a model-free procedure OPEN and presented ways to mitigate its bias and variance. Empirical results suggests that OPEN can now not only enable practitioners to predict future performances amidst non-stationarity but also identify policies that may be actively causing harm or damage. In the future, OPEN can also be extended to enable *control* of non-stationary processes.

## 8    Acknowledgements

Research reported in this paper was sponsored in part by a gift from Adobe, NSF award #2018372. This work was also funded in part by the U.S. Army Combat Capabilities Development Command (DEVCOM) Army Research Laboratory under Cooperative Agreement W911NF-17-2-0196 and Support Agreement No. USMA21050. The views expressed in this paper are those of the authors and do not reflect the official policy or position of the United States Military Academy, the United States Army, the Department of Defense, or the United States Government. The U.S. Government is authorized to reproduce and distribute reprints for Government purposes notwithstanding any copyright notation herein.

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
