# Off-Policy Evaluation for Action-Dependent Non-Stationary Environments
## (Appendix)

## Contents

# A FAQs: Frequently Asked Questions

## A.1 How does the stationarity condition for a time-series differ from that in RL?

Conventionally, stationarity is the time-series literature refers to the condition where the distribution (or few moments) of a finite sub-sequence of random-variables in a time-series remains the same as we shift it along the time index axis [Cox and Miller, 2017]. In contrast, the stationarity condition in the RL setting implies that the environment is fixed [Sutton and Barto, 2018]. This makes the performance $J(\pi)$ of any policy $\pi$ to be a constant value throughout. In this work, we use 'stationarity' as used in the RL literature.

## A.2 Can the POMDP during each episode (Figure 2) itself be non-stationary?

Any source of non-stationarity can be incorporated in the (unobserved) state to induce another stationary POMDP (from which we can obtain a single sequence of interaction). The key step towards tractability is Assumption 1 that enforces additional structure on the performance of any policy across the *sequence* of (non-)stationary POMDPs.

## A.3 What if it is known ahead of time that the non-stationarity is passive only?

In such cases where the underlying changes are independent of the past actions, $\mathbb{E}_{\beta_1}[J_{i+1}(\pi)|J_i(\pi)] = \mathbb{E}_{\beta_2}[J_{i+1}(\pi)|J_i(\pi)]$, for any policies $\beta_1$ and $\beta_2$. Therefore, there is no need for double-counterfactual reasoning to correct for the changes observed in the past. Particularly, in Theorem 1 the second use of importance sampling can be avoided as $\mathbb{E}_{\beta_i,\beta_{i+1}}\left[\rho_i \widehat{J}_{i+1}(\pi)\Big|M_i(\pi)\right] = \mathbb{E}_{\beta_i,\beta_{i+1}}\left[\widehat{J}_{i+1}(\pi)\Big|M_i(\pi)\right]$ under passive non-stationarity. Rest of the procedure for OPEN can be modified accordingly.

## A.4 How should different non-stationarities be treated in the on-policy setting?

Perhaps interestingly, OPEN makes no effective distinction between active and passive non-stationarity in the on-policy setting. Notice that in the on-policy setting, importance ratios $\rho = 1$ everywhere, therefore the use of double counterfactual reasoning has no impact. Intuitively, in the on-policy setting, there is no need to dis-entagle the active and passive sources of non-stationarity, as the prediction needs to be made about the same policy that was used during data collection.

## A.5 Can you tell us more about when would Assumption 1 be (in)valid?

Yes, we provide a detailed discussion on Assumption 1 in Appendix C.

## A.6 What are the limitations and potential negative impacts of the work?

Our work presents the first few steps towards off-policy evaluation in the presence of non-stationarity. Towards this goal, we used Assumption 1 to enforce a higher-order stationarity condition. We have provided extended discussion regarding the same in Appendix C and a practitioner should carefully analyze their problem setup to conclude if the assumption holds (at least approximately).

Further, often off-policy evaluation is used in safety-critical settings, where it is important to provide confidence intervals [Thomas et al., 2015, 2019, Jiang and Huang, 2020]. Because of our use of instrument variables, our estimator may have high-variance. This can be explained by observing the closed form equation in (22) obtained using the IV procedure. Here, $Z$ is the instrument variable and if it is weakly correlated with X (i.e,. $Z^\top X$ has a small magnitude) then $(Z^\top X)^{-1}$ can be large thereby increasing variance. However, our proposed method OPEN only provides point-estimates and thus using it as-is in safety critical settings would be irresponsible.

If the application does exhibit non-stationarity, a practitioner may have to make a tough choice between prior methods that provide confidence intervals under the stationarity assumption, or the proposed method that may be applicable to their non-stationary setting but does not provide any confidence intervals.

# B    Extended Related Work

In this section we discuss several different research directions that are relevant to the topic of this paper. We refer the readers to the work by Padakandla [2020], Khetarpal et al. [2020] for a more exhaustive survey.

## B.1    Off-policy evaluation in stationary domains

In the off-policy RL setup, there is a large body of literature that tackles the off-policy estimation problem. One line of work leverages dynamic programming [Puterman, 1990, Sutton and Barto, 2018] to develop off-policy estimators [Boyan, 1999, Sutton et al., 2008, 2009, Mahmood et al., 2014, 2015]. Several recent approaches also build upon a dual perspective for dynamic programming [Puterman, 1990, Wang et al., 2007, Nachum and Dai, 2020] for performing off-policy evaluation [Liu et al., 2018, Xie et al., 2019, Jiang and Huang, 2020, Uehara et al., 2020, Dai et al., 2020, Feng et al., 2021]. These works require fully-observable states. Other direction of work takes Monte-Carlo perspective to perform trajectory based importance sampling and are applicable to stationary setting with partial observability [Precup, 2000, Thomas et al., 2015, Jiang and Li, 2015, Thomas and Brunskill, 2016]. The proposed work builds upon this direction.

Several works have also discussed various techniques for variance reduction [Jiang and Li, 2015, Thomas and Brunskill, 2016, Munos et al., 2016, Harutyunyan et al., 2016, Liu et al., 2018, Espeholt et al., 2018, Nachum et al., 2019, Yang et al., 2020, Yuan et al., 2021]. However, these methods are restricted to stationary domains.

## B.2    Non-stationarity in stationary domains

In the face of uncertainty, prior works often opt for exploratory or safe behavior by acting optimistically or pessimistically, respectively. This is often achieved by using the collected data to dynamically modify the observed rewards for any state-action pair by either providing bonuses [Agarwal et al., 2020, Taiga et al., 2021] or penalties [Buckman et al., 2020, Cetin and Celiktutan, 2021]. One could view this as an instance of active non-stationarity. Similarly, in temporal-difference (TD) methods the target for the value function keeps changing and such changes are also dependent on the data collected in the past [Sutton and Barto, 2018]. However, we note that such non-stationarities are only artifacts of the learning algorithm as the underlying domain remains stationary throughout. In contrast, the focus of our work is on settings where the underlying domain is non-stationary.

## B.3    Single Episode Continuing setting

As discussed in Section 4, non-stationarity can be alternatively modeled using a single long episode in a stationary POMDP. From this point of view, one may wonder if the average-reward/continuing setting [Sutton and Barto, 2018] could be useful? While there have been off-policy evaluation methods designed to tackle the continuing setting [Liu et al., 2018, Nachum et al., 2019, Yang et al., 2020], they require two important conditions that are no applicable for our setting: **(a)** They assume access to the true underlying state such that there is no partial-observability, and (b) They assume that the transition tuples are sampled from the stationary state-visitation distribution of a policy. In the non-stationary setting that we consider, we may not have data from any stationary state visitation distribution, and we may not have access to the true underlying states either.

## B.4    Non-stationarity in MDPs/Bandits

Several prior methods have considered tackling non-stationarity for reinforcement learning problems. For instance, a Hidden-Mode MDP is a setting that assumes that the environment changes are confined to a few hidden modes, where each mode represents a unique MDP. This provides a tractable way to model a limited number of MDPs [Choi et al., 2000, Basso and Engel, 2009], or perform updates using mode-change detection [Da Silva et al., 2006, Padakandla et al., 2019, Alegre et al., 2021]. Similarly there are methods [Xie et al., 2020a] based on hidden-parameter MDPs [Doshi-Velez and Konidaris, 2016] that consider a more general setup where the hidden variable can be continuous. Alternatively, many methods [Thomas et al., 2017, Jagerman et al., 2019, Chandak et al., 2020b, Zhou et al., 2020, Poiani et al., 2021, Liotet et al., 2021] have considered time-dependent MDPs [Rachelson

et al., 2009]. Aspects related to safety and confidence intervals have also been explored [Ammar et al., 2015, Chandak et al., 2020a, 2021]. However, the focus of these methods are on settings with passive non-stationarity, where the past actions do not influence the underlying non-stationarity. Our works extends this direction of research to provide off-policy evaluation amidst active and hybrid non-stationarity as well.

Non-stationary multi-armed bandits (NMAB) capture the setting where the horizon length is one, but the reward distribution changes over time [Moulines, 2008, Besbes et al., 2014, Russac et al., 2019, Vernade et al., 2020]. Many variants of NMAB, like *cascading non-stationary bandits* [Wang et al., 2019a, Li and de Rijke, 2019] and *rotting bandits* [Levine et al., 2017, Seznec et al., 2018] have also been considered. In contrast, this work focuses on methods that generalize to the sequential decision making setup where the horizon length can be more than 1.

### B.5 Multi-agent Games

Non-stationarity also occurs in multiplayer games [Singh et al., 2000, Bowling, 2005, Conitzer and Sandholm, 2007] where the opponent can change their strategy as a response to the agent's previous decisions. These types of changes are related to active non-stationarity that we consider in this work. In such games, opponent modeling has been shown to be useful and regret bounds for multi-player games [Zhang and Lesser, 2010, Mealing and Shapiro, 2013, Foster et al., 2016, Foerster et al., 2018]. Further, often these games still assume that the underlying system/environment (excluding other players) is stationary and focus on searching for (Nash) equilibria. Similarly, non-stationarities are also induced in the multi-agent systems where an agent tries to influence other agents [Jaques et al., 2019, Wang et al., 2019b, Xie et al., 2020b, Wang et al., 2021]. However, under general non-stationarity, the underlying system may also change and thus there may not even exist any fixed equilibria. Perhaps a more relevant setting would be that of evolutionary/dynamics games, where the pay-off matrix and specification of the game can change over time [Gemp and Mahadevan, 2017, Hennes et al., 2019]. Such methods, however, do not leverage any underlying structure in how the game is changing nor do they account for settings where the changes might be a consequence of past interactions of the agent. While relevant, these other research areas are distinct from our setting of interest.

### B.6 Dynamical Systems and Time-Series Analysis

The proposed method for modeling the evolution of a policy's performance over time using stochastic estimates of past performances may be reminiscent of state-space methods (e.g., Kalman filtering) for dynamical systems [Hamilton, 1994]. However, in comparison to these methods, we do not need to model noise variables, which could have been challenging in our case as noise is heteroskedastic because of past (off-policy) performance estimates being computed using data from different behavior policies. Further the form of OPEN estimator allows leveraging (accelerated) gradient descent based optimizers to obtain the solution instead of relying on computationally expensive closed-form solutions that are typically needed by state-space models. Due to this, in practice our method can also be used with non-linear functions $f$ (e.g., recurrent neural network based auto-regressive models).

Different applications of time-series analysis have also discussed the use of lags as instruments [Achen, 2000, Reed, 2015, Bellemare et al., 2017, Wilkins, 2018, Wang and Bellemare, 2019]. Our use case differs from these prior works in that we look at the full sequential decision making setup for reinforcement learning, and also consider a novel importance-weighted instrument-variable regression model.

## C  Discussion on the Structural Assumption

Assumption 1 states that $\forall m \in \mathcal{M}$ such that the performance $J(\pi)$ associated with $m$ is $j$,

$$\forall i, \ \Pr(J_{i+1}(\pi) = j_{i+1}|M_i = m; \pi) = \Pr(J_{i+1}(\pi) = j_{i+1}|J_i(\pi) = j; \pi).$$

As discussed earlier, consider a 'meta-transition' function that characterizes $\Pr(J_{i+1}(\pi)|J_i(\pi), \pi')$ similar to how the standard transition function in an MDP characterizes $\Pr(S_{t+1}|S_t, A_t)$. This assumption is imposing the following two conditions: **(a)** A *higher-order stationarity* condition on the meta-transitions under which non-stationarity can result in changes over time, but *the way the*

*changes happen is fixed*, and **(b)** Knowing the past performance(s) of a policy $\pi$ provides *sufficient* information for the meta-transition function to model how the performance will change upon executing any (possibly different) policy $\pi'$. We provide some examples in Figure 6 to demonstrate few settings to discuss the applicability of this assumption.

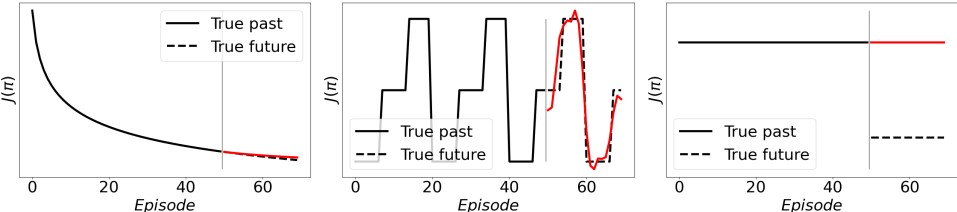

Figure 6: In this figure we plot different kinds of performance trends and discuss the applicability of Assumption 1 for each. The red curve corresponds to the forecast obtained using an auto-regressive model. **(Left)** In many cases where the performance of a policy is smoothly changing over time (for e.g., drifts in interests of an user that a recommender system needs to account for), looking at the past performances can often provide indication of how the performance would evolve in the future. **(Middle)** Changes in performances does not necessarily have to be smooth. What Assumption 1 enforces is that the changes have some structure which can be generalized to make predictions about how the performance would change in the future. Here, the performance jumps between different values (for e.g., if there is discontinuous change in the underlying system), but till their is some structure in the changes, it can be leveraged to make predictions about the future performances as well. **(Right)** While Assumption 1 can be applicable in many setting, there can be settings where this assumption does not hold. For example, if a motor of an industrial system is degrading over time but this degradation has no effect on the observable performance, until the point when the motor breaks down and the performance drops completely. In such cases, just looking at past performances may not be sufficient to infer how performance will change in the future.

## C.1 Latent Variables

Instead of enforcing structure on the performances, a possible alternative could have been to enforce structure on how the underlying latent variable (e.g., friction of a motor, interests of a user) are changing over time. While this might be more intuitive for some, just considering structure on this latent variable need not be sufficient. Dealing with latent/hidden variables can particularly challenging in the off-policy setting, as it may often not be possible (unless additional assumptions are enforced) to infer the latent variable using just the observations from past interactions, even in the stationary setting [Tennenholtz et al., 2020, Namkoong et al., 2020, Shi et al., 2021, Bennett et al., 2021].

Further, the end goal is to estimate the performance of a policy in the future. Therefore, even if we could infer the possible latent variables for the future episodes, it would still require additional regularity conditions on the (unknown) function that maps from the latent variable to the performance associated with it for any given policy. Without that it would not be possible to generalize what would the performance be for the inferred latent variables of the future. And as we discuss in Figure 7, these two assumptions on (a) the structure of how the latent variable could change, and (b) the regularity condition on how the latent variable impacts the performance, can often be reduced to a single condition directly on the structure of how the performances are changing.

# D Proofs for Theoretical Results

## D.1 Double Counterfactual Reasoning

**Theorem 1.** *Under Assumptions 1 and 2, $\forall m \in \mathcal{M}$ such that the performance $J(\pi)$ associated with $m$ is $j$, $\mathbb{E}_\pi \left[ J_{i+1}(\pi) | J_i(\pi) = j \right] = \mathbb{E}_{\beta_i, \beta_{i+1}} \left[ \rho_i \widehat{J}_{i+1}(\pi) \big| M_i = m \right]$.*

*Proof.* In the following, to make the dependence of trajectories explicit, we will additionally define $\rho(h)$ and $g(h)$ to be the importance ratios and the return associated with a trajectory $h$. Using this

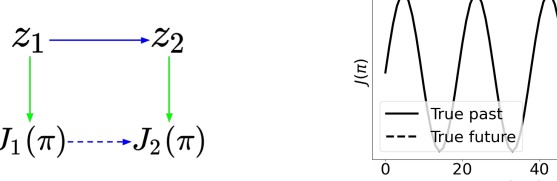

Figure 7: **(Left)** Considering structured changes in latent variable $z$ (blue arrow) of the POMDP might often be more intuitive. However, as $J(\pi)$ estimation is required ultimately, unless performance of a policy also has some structure (green arrows) given $z$, generalizing across (potentially unseen) $z$'s may not be possible. Structured changes for blue and green arrows consequently results in structured changes in $J(\pi)$ (dashed-blue arrows). For example, if the performance $J(\pi)$ of a policy changes (Lipschitz) smoothly with $z$, then (Lipschitz) smooth changes between $z$ values automatically also imply (Lipschitz) smooth changes between $J(\pi)$ values. **(Right)** When executing a policy $\pi$, say $z$ changes as $z_i = i$, and $J_i(\pi)$ changes periodically as $\sin(z_i)$. Here, even though both $z$ and $J$ change smoothly, changes in $z_{i+1}$ can be modeled using one past term (i.e, $z_i$) as given just the current performance value $J_i(\pi)$ it is not possible to predict whether the next performance $J_{i+1}(\pi)$ would increase or decrease in the future. However, such a problem can be easily resolved by looking at multiple past performances to infer the trend (for e.g., just using past 2 terms here suffices to exactly predict the future outcomes(red curve)).

notation, it can be observed that,

$$\mathbb{E}_\pi\left[J_{i+1}(\pi)|M_i\right] = \sum_{h_{i+1}} \Pr(h_{i+1}|M_i;\pi)g(h_{i+1})$$

$$\stackrel{(a)}{=} \sum_{h_{i+1}}\sum_{m_{i+1}}\sum_{h_i} \Pr(h_{i+1},m_{i+1},h_i|M_i;\pi)g(h_{i+1})$$

$$\stackrel{(b)}{=} \sum_{h_i} \Pr(h_i|M_i;\pi) \sum_{m_{i+1}} \Pr(m_{i+1}|h_i,M_i;\pi)$$
$$\sum_{h_{i+1}} \Pr(h_{i+1}|m_{i+1},h_i,M_i;\pi)g(h_{i+1})$$

$$\stackrel{(c)}{=} \sum_{h_i} \Pr(h_i|M_i;\pi) \sum_{m_{i+1}} \Pr(m_{i+1}|h_i,M_i) \sum_{h_{i+1}} \Pr(h_{i+1}|m_{i+1};\pi)g(h_{i+1})$$

$$\stackrel{(d)}{=} \sum_{h_i} \rho(h_i)\Pr(h_i|M_i;\beta_k) \sum_{m_{i+1}} \Pr(m_{i+1}|h_i,M_i)$$
$$\sum_{h_{i+1}} \rho(h_{i+1})\Pr(h_{i+1}|m_{i+1};\beta_{i+1})g(h_{i+1})$$

$$\stackrel{(e)}{=} \sum_{h_i}\sum_{m_{i+1}}\sum_{h_{i+1}} \Pr(h_i|M_i;\beta_i)\Pr(m_{i+1}|h_i,M_i)\Pr(h_{i+1}|m_{i+1};\beta_{i+1})\Big[\rho(h_i)\rho(h_{i+1})g(h_{i+1})\Big]$$

$$= \mathbb{E}_{\beta_i\beta_{i+1}}\left[\rho_i\rho_{i+1}G_{i+1}|M_i\right]$$
$$= \mathbb{E}_{\beta_i\beta_{i+1}}\left[\rho_i\widehat{J}_{i+1}(\pi)|M_i\right], \tag{7}$$

where (a) follows from the law of total probability, (b) follows from the chain rule of probability, (c) follows using conditional independence, where $m_{i+1}$ is independent of $\pi$ given $h_i$ and $M_i$ because of the meta-transition function $\mathcal{T}$, and $h_{i+1}$ i independent of $h_i$ and $M_i$ given $m_{i+1}$ and $\pi$, (d) follows from the use of importance sampling to switch the sampling distribution under Assumption 2, and (e) follows from re-arrangement of terms. Finally, $\rho_i$ and $\rho_{i+1}$ are the random variables corresponding the importance ratios in episodes $i$ and $i+1$. Random variable $G_{i+1}$ corresponds to the return under $\beta$ in episode $i+1$.

Now notice that

$$
\begin{aligned}
\mathbb{E}_\pi\left[J_{i+1}(\pi)|M_i\right] &= \sum_{y\in\mathbb{R}}\Pr(J_{i+1}(\pi)=y|M_i;\pi)y \\
&\stackrel{(f)}{=} \sum_{y\in\mathbb{R}}\Pr(J_{i+1}(\pi)=y|J_i(\pi);\pi)y \\
&= \mathbb{E}_\pi\left[J_{i+1}(\pi)|J_i(\pi)\right],
\end{aligned}
\tag{8}
$$

where $(f)$ follows from Assumption 1. Finally, combining (7) and (8),

$$
\mathbb{E}_\pi\left[J_{i+1}(\pi)|J_i(\pi)\right] = \mathbb{E}_{\beta_i,\beta_{i+1}}\left[\rho_i\widehat{J}_{i+1}(\pi)\Big|M_i\right].
$$

$\square$

Similarly, under a more generalized Assumption 1, where $\forall m\in\mathcal{M}$,

$$
\forall i>p,\ \Pr(J_{i+1}(\pi)=j_{i+1}|M_i=m;\pi') = \Pr(J_{i+1}(\pi)=j_{i+1}|(J_{i-k}(\pi)=j_{i-k})_{k=0}^p;\pi').
$$

then similar steps as earlier can be used to conclude that

$$
\mathbb{E}_\pi\left[J_{i+1}(\pi)|(J_{i-k}(\pi))_{k=0}^p\right] = \mathbb{E}_{\beta_i\beta_{i+1}}\left[\rho_i\widehat{J}_{i+1}(\pi)|M_i\right].
\tag{9}
$$

Note that no additional importance correction is needed in (9) compared to (8). The term $\rho_i$ only shows up to correct for the transition between $M_i$ and $M_{i+1}$ due to the meta-transition function $\mathcal{T}(m,h,m') = \Pr(M_{i+1}=m'|M_i=m, H_i=h)$. This independence on the choice of $p$ also holds if $\mathcal{T}$ is non-Markovian in the previous $M_i$ values. Although, additional importance correction would be required if $\mathcal{T}$ is dependent on multiple past $H_i$ terms.

### D.2  Asymptotic bias of $\hat{\theta}_{naive}$

Recall that $\hat{\theta}_{\text{naive}}$ is given by,

$$
\hat{\theta}_{\text{naive}} \in \underset{\theta\in\Theta}{\operatorname{argmin}} \sum_{i=1}^{n-1}\left(f\left(\widehat{J}_i(\pi);\theta\right) - \rho_i\widehat{J}_{i+1}(\pi)\right)^2.
$$

Because $\widehat{J}_i(\pi)$ is an unbiased estimate of $J_\pi$, let $\widehat{J}_i(\pi) = J_i(\pi) + \eta_i$, where $\eta_i$ is a mean zero noise. Let $\mathbb{N} := [\eta_1, \eta_2, ..., \eta_{n-1}]^\top$ and $\mathbb{J} := [J_1(\pi), J_2(\pi), ..., J_{n-1}(\pi)]^\top$. When $f$ is a linear function of its inputs, expected value $\mathbb{E}_\pi[J_{i+1}(\pi)|J_i(\pi)] = J_i\theta_\pi$. Also, as $\rho_i\hat{J}_{i+1}(\pi)$ is an unbiased estimator for $J_i(\pi)\theta_\pi$ given $J_i(\pi)$, let $\rho_i\hat{J}_{i+1}(\pi) = J_i(\pi)\theta_\pi + \zeta_i$, where $\zeta_i$ is mean zero noise. Let $\mathbb{N}_2 := [\zeta_1, \zeta_2, ..., \zeta_{n-1}]^\top$ then $\theta_{\text{naive}}$ can be expressed as,

$$
\begin{aligned}
\hat{\theta}_{\text{naive}} &= \left((\mathbb{J}+\mathbb{N})^\top(\mathbb{J}+\mathbb{N})\right)^{-1}(\mathbb{J}+\mathbb{N})^\top(\mathbb{J}\theta_\pi+\mathbb{N}_2) \\
&= \left(\mathbb{J}^\top\mathbb{J}+2\mathbb{J}^\top\mathbb{N}+\mathbb{N}^\top\mathbb{N}\right)^{-1}\left(\mathbb{J}^\top\mathbb{J}\theta_\pi+\mathbb{N}^\top\mathbb{J}\theta_\pi+\mathbb{J}^\top\mathbb{N}_2+\mathbb{N}^\top\mathbb{N}_2\right) \\
&= \left(\frac{1}{n}\left(\mathbb{J}^\top\mathbb{J}+2\mathbb{J}^\top\mathbb{N}+\mathbb{N}^\top\mathbb{N}\right)\right)^{-1}\left(\frac{1}{n}\left(\mathbb{J}^\top\mathbb{J}\theta_\pi+\mathbb{N}^\top\mathbb{J}\theta_\pi+\mathbb{J}^\top\mathbb{N}_2+\mathbb{N}^\top\mathbb{N}_2\right)\right).
\end{aligned}
\tag{10}
$$

In the limit, using continuous mapping theorem when the inverse in (10) exists,

$$
\lim_{n\to\infty}\hat{\theta}_{\text{naive}} = \left(\lim_{n\to\infty}\frac{1}{n}\left(\mathbb{J}^\top\mathbb{J}+2\mathbb{J}^\top\mathbb{N}+\mathbb{N}^\top\mathbb{N}\right)\right)^{-1}\left(\lim_{n\to\infty}\frac{1}{n}\left(\mathbb{J}^\top\mathbb{J}\theta_\pi+\mathbb{N}^\top\mathbb{J}\theta_\pi+\mathbb{J}^\top\mathbb{N}_2+\mathbb{N}^\top\mathbb{N}_2\right)\right).
\tag{11}
$$

Observe that both $\mathbb{N}$ and $\mathbb{N}_2$ are mean zero and uncorrelated with each other and also with $\mathbb{J}$. Therefore, the terms corresponding to $\mathbb{J}^\top\mathbb{N}$, $\mathbb{J}^\top\mathbb{N}_2$, and $\mathbb{N}^\top\mathbb{N}_2$ in (11) will be zero almost surely due to Rajchaman's strong law of large numbers for uncorrelated random variables [Rajchman, 1932, Chandra, 1991]. However, the term corresponding to $\mathbb{N}^\top\mathbb{N}$ will not be zero in the limit, and instead roughly result in (average of the) variances of $\eta_i$. Consequently, this results in,

$$
\hat{\theta}_{\text{naive}} \xrightarrow{a.s.} \left(\mathbb{J}^\top\mathbb{J}+\mathbb{N}^\top\mathbb{N}\right)^{-1}\mathbb{J}^\top\mathbb{J}\theta_\pi.
$$

### D.3 Importance-Weighted IV-Regression

**Theorem 2.** *Under Assumptions* 1 *and* 2, $\forall i,$ $\quad \text{Cov}\left(\widehat{J}_{i-1}(\pi), \widehat{J}_i(\pi) - J_i(\pi)\right) = 0.$

*Proof.*

$$\forall i, \text{Cov}\left(\widehat{J}_i(\pi), \widehat{J}_{i+1}(\pi) - J_{i+1}(\pi)\right) = \underbrace{\mathbb{E}_\beta\left[\widehat{J}_i(\pi)\left(\widehat{J}_{i+1}(\pi) - J_{i+1}(\pi)\right)\right]}_{\text{(I)}}$$

$$- \underbrace{\mathbb{E}_\beta\left[\widehat{J}_i(\pi)\right]\mathbb{E}_\beta\left[\widehat{J}_{i+1}(\pi) - J_{i+1}(\pi)\right]}_{\text{(II)}}. \qquad (12)$$

Focusing on term (II),

$$\mathbb{E}_\beta\left[\widehat{J}_i(\pi)\right]\mathbb{E}_\beta\left[\widehat{J}_{i+1}(\pi) - J_{i+1}(\pi)\right] = \mathbb{E}_\beta\left[\widehat{J}_i(\pi)\right]\left(\mathbb{E}_\beta\left[\widehat{J}_{i+1}(\pi)\right] - J_{i+1}(\pi)\right)$$

$$\overset{(a)}{=} \mathbb{E}_\beta\left[\widehat{J}_i(\pi)\right]\left(J_{i+1}(\pi) - J_{i+1}(\pi)\right)$$

$$= 0,$$

where (a) follows from the fact that under Assumption 2, $\widehat{J}_{i+1}(\pi)$ is an unbiased estimator for $J_{i+1}(\pi)$ [Thomas, 2015]. Focusing on term (I) and using the law of total expectation,

$$\mathbb{E}_\beta\left[\widehat{J}_i(\pi)\left(\widehat{J}_{i+1}(\pi) - J_{i+1}(\pi)\right)\right] = \mathbb{E}_\beta\left[\widehat{J}_i(\pi)\underbrace{\mathbb{E}_\beta\left[\widehat{J}_{i+1}(\pi) - J_{i+1}(\pi)\Big|\widehat{J}_i(\pi)\right]}_{\text{(III)}}\right].$$

Expanding term (III) further using the law of total expectation,

$$\mathbb{E}_\beta\left[\widehat{J}_{i+1}(\pi) - J_{i+1}(\pi)\Big|\widehat{J}_i(\pi)\right] \overset{(b)}{=} \mathbb{E}_\beta\left[\mathbb{E}_\beta\left[\widehat{J}_{i+1}(\pi) - J_{i+1}(\pi)\Big|M_{i+1}, \widehat{J}_i(\pi)\right]\Big|\widehat{J}_i(\pi)\right]$$

$$\overset{(c)}{=} \mathbb{E}_\beta\left[\mathbb{E}_\beta\left[\widehat{J}_{i+1}(\pi) - J_{i+1}(\pi)\Big|M_{i+1}\right]\Big|\widehat{J}_i(\pi)\right]$$

$$\overset{(d)}{=} 0,$$

where in (b) the outer expectation is over the next environment $M_{i+1}$ given that the current performance estimate is $\widehat{J}_i(\pi)$ and that $\beta_i$ was used for interaction in episode $i$. The inner expectation is over $\widehat{J}_{i+1}(\pi)$, where the trajectory used for estimating $\widehat{J}_{i+1}(\pi)$ is collected using $\beta$ in the environment $M_{i+1}$. Step (c) follows from the fact that conditioned on the environment $M_{i+1}$, interactions in $M_{i+1}$ are independent of quantities observed in the episodes before $i+1$. Finally, step (d) follows from observing that

$$\mathbb{E}_\beta\left[\widehat{J}_{i+1}(\pi) - J_{i+1}(\pi)\Big|M_{i+1}\right] = \mathbb{E}_\beta\left[\widehat{J}_{i+1}(\pi)\Big|M_{i+1}\right] - J_{i+1}(\pi)$$

$$\overset{(e)}{=} J_{i+1}(\pi) - J_{i+1}(\pi)$$

$$= 0,$$

where (e) follows from the fact that under Assumption 2, $\widehat{J}_{i+1}(\pi)$ is an unbiased estimator of the performance of $\pi$ for the given environment $M_{i+1}$. Therefore both (a) and (b) in (12) are zero, and we conclude the result. $\qquad \square$

**Theorem 3.** *Under Assumptions* 1, 2, *and* 3, *if* $f$ *and* $g$ *are linear functions of their inputs, then* $\hat{\theta}_n$ *is a strongly consistent estimator of* $\theta_\pi$, *i.e.,* $\hat{\theta}_n \overset{a.s.}{\longrightarrow} \theta_\pi$. *(See Appendix D.3 for the proof.)*

*Proof.* For the linear setting, $\hat{\theta}_n$ can be expressed as,

$$\hat{\phi}_n \in \underset{\phi\in\Phi}{\text{argmin}} \sum_{i=2}^{n-1}\left(\widehat{J}_{i-1}(\pi)\phi - \widehat{J}_i(\pi)\right)^2. \qquad (13)$$

$$\hat{\theta}_n \in \underset{\theta\in\Theta}{\text{argmin}} \sum_{i=2}^{n-1}\left(\bar{J}_i(\pi)\theta - \rho_i\widehat{J}_{i+1}(\pi)\right)^2, \qquad \text{where} \quad \bar{J}_i := \widehat{J}_{i-1}\hat{\phi}_n. \qquad (14)$$

Before moving further, we introduce some additional notations. Particularly, we will use matrix based notations such that it provides more insights into how the steps would work out for other choices of instrument variables as well.

$$\mathbf{X_1} := \left[\widehat{J}_1(\pi), ..., \widehat{J}_{n-2}(\pi)\right]^\top, \qquad \mathbf{\Lambda_1} := \texttt{diag}([\rho_1, ..., \rho_{n-2}]),$$

$$\mathbf{X_2} := \left[\widehat{J}_2(\pi), ..., \widehat{J}_{n-1}(\pi)\right]^\top, \qquad \mathbf{\Lambda_2} := \texttt{diag}\left([\rho_2, ..., \rho_{n-1}]\right),$$

$$\mathbf{X_3} := \left[\widehat{J}_3(\pi), ..., \widehat{J}_n(\pi)\right]^\top \qquad \mathbf{\bar{X}_2} := \left[\bar{J}_2(\pi), ..., \bar{J}_{n-1}(\pi)\right]^\top,$$

where the $\texttt{diag}$ corresponds to a diagonal matrix with off-diagonals set to zero.

In the following, we split the proof in two parts: (a) we will first show that

$$\hat{\theta}_n = \left(\mathbf{X_1^\top X_2}\right)^{-1}\left(\mathbf{X_1^\top \Lambda_2 X_3}\right),$$

and then (b) using this simplified form for $\hat{\theta}_n$ we will show that $\hat{\theta}_n \xrightarrow{\text{a.s.}} \theta_\pi$.

**Part (a)**  Solving (13) in matrix form,

$$\hat{\phi}_n = \left(\mathbf{X_1^\top X_1}\right)^{-1}\mathbf{X_1^\top X_2}. \tag{15}$$

Similarly, solving (14) in matrix form,

$$\hat{\theta}_n = \left(\mathbf{\bar{X}_2^\top \bar{X}_2}\right)^{-1}\mathbf{\bar{X}_2^\top \Lambda_2 X_3}. \tag{16}$$

Now substituting the value of $\mathbf{\bar{X}_2}$ in (16),

$$\hat{\theta}_n = \left(\left(\underbrace{\mathbf{X_1}\hat{\phi}_\mathbf{n}}_{\mathbf{\bar{X}_2}}\right)^\top \left(\underbrace{\mathbf{X_1}\hat{\phi}_\mathbf{n}}_{\mathbf{\bar{X}_2}}\right)\right)^{-1}\left(\underbrace{\mathbf{X_1}\hat{\phi}_\mathbf{n}}_{\mathbf{\bar{X}_2}}\right)^\top \mathbf{\Lambda_2 X_3}. \tag{17}$$

Using (15) to substitute the value of $\hat{\phi}_n$ in (17),

$$\hat{\theta}_n = \left(\left(\mathbf{X_1}\underbrace{\left(\mathbf{X_1^\top X_1}\right)^{-1}\mathbf{X_1^\top X_2}}_{\hat{\phi}_\mathbf{n}}\right)^\top \left(\mathbf{X_1}\underbrace{\left(\mathbf{X_1^\top X_1}\right)^{-1}\mathbf{X_1^\top X_2}}_{\hat{\phi}_\mathbf{n}}\right)\right)^{-1}$$
$$\left(\mathbf{X_1}\underbrace{\left(\mathbf{X_1^\top X_1}\right)^{-1}\mathbf{X_1^\top X_2}}_{\hat{\phi}_\mathbf{n}}\right)^\top \mathbf{\Lambda_2 X_3}. \tag{18}$$

Using matrix operations to expand the transposes in (18),

$$\hat{\theta}_n = \left(\left(\mathbf{X_2^\top X_1}\left(\mathbf{X_1^\top X_1}\right)^{-1}\mathbf{X_1^\top}\right)\left(\mathbf{X_1}\left(\mathbf{X_1^\top X_1}\right)^{-1}\mathbf{X_1^\top X_2}\right)\right)^{-1}$$
$$\left(\mathbf{X_2^\top X_1}\left(\mathbf{X_1^\top X_1}\right)^{-1}\mathbf{X_1^\top}\right)\mathbf{\Lambda_2 X_3}. \tag{19}$$

Similarly, using matrix operations to expand inverses in (19) (colored underlines are used to match the terms before expansion in (19) and after expansion in (20)),

$$\hat{\theta}_n = \left(\mathbf{X_1^\top X_2}\right)^{-1}\left(\mathbf{X_1^\top X_1}\right)\left(\mathbf{X_1^\top X_1}\right)^{-1}\left(\mathbf{X_1^\top X_1}\right)\left(\mathbf{X_2^\top X_1}\right)^{-1}$$
$$\left(\mathbf{X_2^\top X_1}\right)\left(\mathbf{X_1^\top X_1}\right)^{-1}\left(\mathbf{X_1^\top \Lambda_2 X_3}\right), \tag{20}$$

Notice that several terms in (20) cancel each other out, therefore,

$$\hat{\theta}_n = \left(\mathbf{X_1^\top X_2}\right)^{-1}\left(\mathbf{X_1^\top \Lambda_2 X_3}\right). \tag{21}$$

As a side remark, we note that if we replace $\mathbf{X_1}$ in the above steps with an appropriate instrument variable $\mathbf{Z_1}$, then similar steps will follow and will result in

$$\hat{\theta}_n = \left(\mathbf{Z_1^\top X_2}\right)^{-1}\left(\mathbf{Z_1^\top \Lambda_2 X_3}\right). \tag{22}$$

**Part (b)** Now when $f(J_i(\pi); \theta_\pi) := \mathbb{E}_\pi [J_{i+1}(\pi) | J_i(\pi)]$ is a linear function,

$$J_{i+1}(\pi) = J_i(\pi)\theta_\pi + U_{i+1}(H_i),$$

where $U_{i+1}$ is a bounded mean zero noise (which depends on the interaction $H_i$ by $\pi$). Using Theorem 1, let

$$Y_{i+1} := \mathbb{E}_\pi [J_{i+1}(\pi) | J_i(\pi)]$$

and its unbiased estimate be

$$\widehat{Y}_{i+1} := \rho_i \widehat{J}_{i+1}(\pi) = \rho_i \rho_{i+1} G_{i+1}. \tag{23}$$

For the regression, since $\widehat{J}_i(\pi)$ is an unbiased estimate of the input $J_i(\pi)$ and $\widehat{Y}_{i+1}$ is an unbiased estimate of the target $\mathbb{E}_\pi [J_{i+1}(\pi) | J_i(\pi)]$, these can be equivalently expressed as,

$$\widehat{J}_i(\pi) = J_i(\pi) + V_i(H_i),$$
$$\widehat{Y}_{i+1} = J_{i+1}(\pi) + W_{i+1}(H_i, H_{i+1}), \tag{24}$$

where $V_i(H_i)$ is some bounded mean-zero noise (dependent on the unbiased estimate made using $H_i$) and $W_{i+1}(H_i, H_{i+1})$ is also a bounded mean-zero noise (dependent on the unbiased estimate made using $H_i$ and $H_{i+1}$). Before moving further, we define some additional notation,

$$\begin{aligned} \mathbf{Y_3} &:= [Y_3, ..., Y_n]^\top & \mathbf{U_3} &:= [U_3(H_2), ..., U_n(H_{n-1})]^\top, \\ \widehat{\mathbf{Y}}_3 &:= [\widehat{Y}_3, ..., \widehat{Y}_n]^\top & \mathbf{V_2} &:= [V_2(H_2), ..., V_{n-1}(H_{n-1})]^\top. \\ \mathbb{J}_2 &:= [J_2(\pi), ..., J_{n-1}(\pi)]^\top & \mathbf{W_3} &:= [W_3(H_2, H_3), ..., W_n(H_{n-1}, H_n)]^\top. \end{aligned}$$

Using (23) note that $\widehat{\mathbf{Y}}_3 = \mathbf{\Lambda_2 X_3}$, therefore (21) can be expressed as,

$$\hat{\theta}_n = \left(\mathbf{X_1^\top X_2}\right)^{-1} \left(\mathbf{X_1^\top \widehat{Y}_3}\right). \tag{25}$$

Unrolling value of $\widehat{\mathbf{Y}}_3$ in (25) using relations from (23) and (24),

$$\begin{aligned} \hat{\theta}_n &= \left(\mathbf{X_1^\top X_2}\right)^{-1} \left(\mathbf{X_1^\top} \left(\mathbf{Y_3 + W_3}\right)\right) \\ &= \left(\mathbf{X_1^\top X_2}\right)^{-1} \left(\mathbf{X_1^\top} \left(\mathbb{J}_2 \theta_\pi + \mathbf{U_3 + W_3}\right)\right) \\ &= \left(\mathbf{X_1^\top X_2}\right)^{-1} \left(\mathbf{X_1^\top} \left(\left(\mathbf{X_2 - V_2}\right)\theta_\pi + \mathbf{U_3 + W_3}\right)\right). \end{aligned} \tag{26}$$

Expanding (26),

$$\hat{\theta}_n = \theta_\pi - \left(\mathbf{X_1^\top X_2}\right)^{-1} \mathbf{X_1^\top V_2}\theta_\pi + \left(\mathbf{X_1^\top X_2}\right)^{-1} \left(\mathbf{X_1^\top} \left(\mathbf{U_3 + W_3}\right)\right). \tag{27}$$

Evaluating the value of (27) in the limit,

$$\lim_{n\to\infty} \hat{\theta}_n = \theta_\pi - \lim_{n\to\infty} \left( \underbrace{\left(\mathbf{X_1^\top X_2}\right)^{-1} \mathbf{X_1^\top V_2}\theta_\pi}_{(a)} + \underbrace{\left(\mathbf{X_1^\top X_2}\right)^{-1} \left(\mathbf{X_1^\top} \left(\mathbf{U_3 + W_3}\right)\right)}_{(b)} \right). \tag{28}$$

It can be now seen from (28) that if in the limit the terms inside the paranthesis are zero, then we would obtain our desired result. Focusing on the term (a) and using the continuous mapping theorem,

$$\begin{aligned} \lim_{n\to\infty} \left(\mathbf{X_1^\top X_2}\right)^{-1} \mathbf{X_1^\top V_2}\theta_\pi &= \lim_{n\to\infty} \left(\frac{1}{n}\mathbf{X_1^\top X_2}\right)^{-1} \left(\frac{1}{n}\mathbf{X_1^\top V_2}\theta_\pi\right) \\ &= \left(\lim_{n\to\infty} \frac{1}{n}\mathbf{X_1^\top X_2}\right)^{-1} \left(\underbrace{\lim_{n\to\infty} \frac{1}{n}\mathbf{X_1^\top V_2}}_{(c)}\right)\theta_\pi, \end{aligned} \tag{29}$$

where Assumption 3 ensures that $\mathbf{X_1}$ and $\mathbf{X_2}$ are correlated and thus their dot product is not zero. Notice that term (c) (29) can be expressed as $\frac{1}{n} \sum_{i=2}^{n-1} X_{i-1} V_i$. Further, recall from Theorem 2 that $V_i$ is a mean zero random variable uncorrelated with $X_{i-1}$ for all $i$. Further, $V_i$ and $X_{i-1}$ are also

bounded for all $i$ as both rewards and importance ratios are bounded (Assumption 2), and $T$ is finite. Now, for $\alpha_i := X_{i-1}V_i$ observe that $\mathbb{E}[\alpha_i] = \mathbb{E}[X_{i-1}\mathbb{E}[V_i|X_{i-1}]] = \mathbb{E}[X_{i-1}0] = 0$ and thus $\alpha_i$ is a bounded and mean zero random variable $\forall i$. Therefore, as $(c)$ is an average of $\alpha$ variables, it follows from the Rajchaman's strong law of large numbers for uncorrelated random variables [Rajchman, 1932, Chandra, 1991] that term under $(c)$ is zero almost surely. Thus,

$$\left(\mathbf{X_1^\top X_2}\right)^{-1}\mathbf{X_1^\top V_2}\theta_\pi \xrightarrow{a.s.} \mathbf{0}.$$

Similarly, for term (b) in (28) observe that both $\mathbf{U_3}$ and $\mathbf{W_3}$ are zero mean random variables uncorrelated with $\mathbf{X_1}$. Therefore, term (b) in (28) is also zero in the limit almost surely. It can now be concluded from (28) that

$$\hat{\theta}_n \xrightarrow{a.s.} \theta_\pi.$$

$\square$

**Theorem 4.** *Under Assumptions 1, 2, and 3, if $f$ and $g$ are linear functions of their inputs, then $\tilde{\theta}_n$ is a strongly consistent estimator of $\theta_\pi$, i.e., $\tilde{\theta}_n \xrightarrow{a.s.} \theta_\pi$. (See Appendix D.3 for the proof.)*

*Proof.* For the linear setting, $\tilde{\theta}_n$ can be expressed as,

$$\hat{\phi}_n \in \operatorname*{argmin}_{\phi\in\Phi} \sum_{i=2}^{n-1} \rho_i \left(\widehat{J}_{i-1}(\pi)\phi - G_i(\pi)\right)^2. \tag{30}$$

$$\tilde{\theta}_n \in \operatorname*{argmin}_{\theta\in\Theta} \sum_{i=2}^{n-1} \rho_i\rho_{i+1} \left(\bar{J}_i(\pi)\theta - G_{i+1}(\pi)\right)^2, \qquad \text{where} \quad \bar{J}_i := \widehat{J}_{i-1}\hat{\phi}_n. \tag{31}$$

Notice that as dividing the objective by a positive constant does not change the optima, we drop the denominator terms in

$$\bar{\rho}_i := \frac{\rho_i\rho_{i+1}}{\sum_{j=2}^{n-1}\rho_j\rho_{j+1}}$$

for the purpose of the analysis. Before moving further, we introduce some additional notations besides the ones introduced in the proof of Theorem 3,

$$\mathbf{G_3} := [G_3,...,G_n]^\top \qquad\qquad \bar{\mathbf{\Lambda}}_\mathbf{2} := \mathtt{diag}([\rho_2\rho_3, \rho_3\rho_4..., \rho_{n-1}\rho_n]),$$

Solving (30) in matrix form,

$$\hat{\phi}_n = \left(\mathbf{X_1^\top \Lambda_2 X_1}\right)^{-1}\mathbf{X_1^\top \Lambda_2 G_2}.$$
$$= \left(\mathbf{X_1^\top \Lambda_2 X_1}\right)^{-1}\mathbf{X_1^\top X_2}.$$

Similarly, solving (31) in matrix form,

$$\tilde{\theta}_n = \left(\bar{\mathbf{X}}_\mathbf{2}^\top \bar{\mathbf{\Lambda}}_\mathbf{2}\bar{\mathbf{X}}_\mathbf{2}\right)^{-1}\bar{\mathbf{X}}_\mathbf{2}^\top \bar{\mathbf{\Lambda}}_\mathbf{2}\mathbf{G_3}.$$
$$\stackrel{(a)}{=} \left(\bar{\mathbf{X}}_\mathbf{2}^\top \bar{\mathbf{\Lambda}}_\mathbf{2}\bar{\mathbf{X}}_\mathbf{2}\right)^{-1}\bar{\mathbf{X}}_\mathbf{2}^\top \mathbf{\Lambda_2 X_3}, \tag{32}$$

where (a) follows from the fact that $\rho_i\rho_{i+1}G_{i+1} = \rho_i\widehat{J}_{i+1}(\pi)$. Now substituting the value of $\bar{\mathbf{X}}_\mathbf{2}$ in (32) similar to (17) and (18) in the proof of Theorem 3,

$$\tilde{\theta}_n = \left(\left(\mathbf{X_2^\top X_1}\left(\mathbf{X_1^\top \Lambda_2 X_1}\right)^{-1}\mathbf{X_1^\top}\right)\bar{\mathbf{\Lambda}}_\mathbf{2}\left(\mathbf{X_1}\left(\mathbf{X_1^\top \Lambda_2 X_1}\right)^{-1}\mathbf{X_1^\top X_2}\right)\right)^{-1}$$
$$\left(\mathbf{X_2^\top X_1}\left(\mathbf{X_1^\top X_1}\right)^{-1}\mathbf{X_1^\top}\right)\mathbf{\Lambda_2 X_3}. \tag{33}$$

Similarly, using matrix operations to expand inverses in (33) (colored underlines are used to match the terms before expansion in (33) and after expansion in (34)) and multiplying and dividing by $n$,

$$\tilde{\theta}_n = \left(\mathbf{X_1^\top X_2}\right)^{-1}\left(\frac{1}{n}\mathbf{X_1^\top \Lambda_2 X_1}\right)\left(\frac{1}{n}\mathbf{X_1^\top \bar{\Lambda}_2 X_1}\right)^{-1}\left(\frac{1}{n}\mathbf{X_1^\top \Lambda_2 X_1}\right)$$
$$\left(\mathbf{X_2^\top X_1}\right)^{-1} \quad \left(\mathbf{X_2^\top X_1}\right)\left(\frac{1}{n}\mathbf{X_1^\top \Lambda_2 X_1}\right)^{-1}\left(\mathbf{X_1^\top \Lambda_2 X_3}\right). \tag{34}$$

Now focusing on the term underlined in green, in the limit,

$$\lim_{n\to\infty} \frac{1}{n}\mathbf{X_1^\top \bar{\Lambda}_2 X_1} = \lim_{n\to\infty} \frac{1}{n} \sum_{i=2}^{n-1} \rho_i \rho_{i+1} \widehat{J}_{i-1}(\pi)\widehat{J}_{i-1}(\pi)^\top$$

$$\overset{(a)}{=} \lim_{n\to\infty} \frac{1}{n} \sum_{i=2}^{n-1} \mathbb{E}_{\beta_i,\beta_{i+1}}[\rho_i \rho_{i+1}] \widehat{J}_{i-1}(\pi)\widehat{J}_{i-1}(\pi)^\top + \frac{1}{n} \sum_{i=2}^{n-1} \varepsilon_i \widehat{J}_{i-1}(\pi)\widehat{J}_{i-1}(\pi)^\top$$

$$\overset{(b)}{=} \lim_{n\to\infty} \frac{1}{n} \sum_{i=2}^{n-1} \widehat{J}_{i-1}(\pi)\widehat{J}_{i-1}(\pi)^\top$$

$$= \lim_{n\to\infty} \frac{1}{n}\mathbf{X_1^\top X_1}, \tag{35}$$

where in (a) we defined random variable $\rho_i \rho_{i+1}$ as its expected value $E_{\beta_i,\beta_{i+1}}[\rho_i\rho_{i+1}]$ plus a mean zero noise $\varepsilon_i$. Step (b) follows from first observing that $\rho_i$ and $\rho_{i+1}$ are uncorrelated. Therefore $\mathbb{E}_{\beta_i,\beta_{i+1}}[\rho_i\rho_{i+1}] = \mathbb{E}_{\beta_i}[\rho_i]\mathbb{E}_{\beta_{i+1}}[\rho_{i+1}] = 1$ as the expected value of importance ratios is 1 [Thomas, 2015]. Similarly, $\varepsilon_i$ is uncorrelated with $\widehat{J}_{i-1}(\pi)$, i.e., the expected value $\mathbb{E}_{\beta_i,\beta_{i+1}}\left[\varepsilon_i | \widehat{J}_{i-1}(\pi)\right] = \mathbb{E}_{\beta_i,\beta_{i+1}}[\varepsilon_i] = 0$ for any given $J_{i-1}(\pi)$. (Intuitively, this step can be seen analogous to the derivation of PDIS, where the expected value of future IS ratios is always one, irrespective of the past events that it has been conditioned on). Now notice that the random variable $\zeta_i := \varepsilon_i\widehat{J}_{i-1}(\pi)\widehat{J}_{i-1}(\pi)^\top$ is bounded and has mean zero for all $i$. Therefore, while $\zeta_i$ and $\zeta_j$ may be dependent, they are uncorrelated for all $i \neq j$. Using strong law of large number for uncorrelated random variables [Rajchman, 1932, Chandra, 1991] the second term in (a) is zero almost surely.

Similarly, it can be observed that $\frac{1}{n}\mathbf{X_1^\top \Lambda_2 X_1}$ converges to $\frac{1}{n}\mathbf{X_1^\top X_1}$. Therefore using (35) in (34), and using the continuous mapping theorem,

$$\tilde{\theta}_n \overset{a.s.}{\longrightarrow} \left(\mathbf{X_1^\top X_2}\right)^{-1} \left(\frac{1}{n}\mathbf{X_1^\top X_1}\right) \left(\frac{1}{n}\mathbf{X_1^\top X_1}\right)^{-1} \left(\frac{1}{n}\mathbf{X_1^\top X_1}\right) \left(\mathbf{X_2^\top X_1}\right)^{-1}$$

$$\left(\mathbf{X_2^\top X_1}\right) \left(\frac{1}{n}\mathbf{X_1^\top X_1}\right)^{-1} \left(\mathbf{X_1^\top \Lambda_2 X_3}\right). \tag{36}$$

Notice that Assumption 3 ensures that $\mathbf{X_1}$ and $\mathbf{X_2}$ are correlated and thus their dot product is not zero. Further, several terms in (36) cancel each other out, therefore,

$$\tilde{\theta}_n \overset{a.s.}{\longrightarrow} \left(\mathbf{X_1^\top X_2}\right)^{-1} \left(\mathbf{X_1^\top \Lambda_2 X_3}\right).$$

Now proof can be completed similarly to the part (b) of the proof of Theorem 3. □

# E   Empirical Details

The code for all the algorithms and experiments can be found here https://github.com/yashchandak/activeNS

## E.1   Algorithm

In Section 5 we established the key insight for how to forecast the next performance based on a single previous performance, when the true performance trend of a policy can be modeled auto-regressively using a single past term. However, as noted in Remark 3 and Figure 7 using more terms can provide more flexibility in the the type of trends that can be modeled. Therefore, we leverage statistics based on multiple past terms to form the instrument variable $Z_i$.

One immediate choice for $Z_i$ is $\widehat{J}_i(\pi)$. However, we found that the high variance of IS estimate makes $\widehat{J}_i(\pi)$ a weak instrument variable [Pearl et al., 2000], that is not strongly correlated with $J_{i+1}(\pi)$. Better choices of $Z_i$ may be the ones that are strongly correlated with $J_{i+1}(\pi)$ but uncorrelated with the noise in the $\widehat{J}_{i+1}(\pi)$ estimate. We found that an alternate choice of $Z_i$ composed of the unweighted return $G_i$ and a WIS-like estimate for $J_i(\pi)$ (where the normalization is done only using

the importance ratios from episodes before $i$) to be more useful. Specifically, we let $Z_i \coloneqq [G_i, \widetilde{J}_i(\pi)]$, where

$$\widetilde{J}_i(\pi) \coloneqq \frac{\rho_i G_i}{\frac{n}{i} \sum_{k=1}^{i} \rho_k}.$$

It can be observed similar to Theorem 2 that this $Z_i$ is uncorrelated with the noise in $\widehat{J}_{i+1}(\pi)$ as well. Further, the weighted version $\widetilde{J}_i(\pi)$ suffers less from variance and we found it to be more strongly correlated with $J_{i+1}(\pi)$. Further, often the performance of the behavior policy is positively/negatively correlated with the performance of the evaluation policy and thus $G_i$ tends to be correlated with $J_{i+1}(\pi)$ as well. One could also explore other potential IVs; we leave this for future work.

Now using past $p$ values of $Z_i$ to form the complete instrument variable, where $p$ is a hyper-parameter, we use the following importance weighted instrument-variable regression,

$$\tilde{\varphi}_n \in \operatorname*{argmin}_{\varphi \in \Omega} \sum_{i=p+1}^{n} \bar{\rho}_i \left( g \left( (Z_j(\pi))_{j=i-p}^{i-1} ; \varphi \right) - G_i(\pi) \right)^2,$$

$$\tilde{\theta}_n \in \operatorname*{argmin}_{\theta \in \Theta} \sum_{i=2p}^{n-1} \rho_i^\dagger \left( f \left( (\bar{J}_j(\pi))_{j=i-p+1}^{i} ; \theta \right) - G_{i+1}(\pi) \right)^2,$$

where,

$$\bar{J}_i(\pi) = g \left( (Z_j(\pi))_{j=i-p}^{i-1} ; \tilde{\varphi}_n \right), \quad \forall p < i \le n,$$

$$\bar{\rho}_i \coloneqq \frac{\rho_i}{\left( \sum_{j=2}^{n} \rho_j \right)}$$

$$\rho_i^\dagger \coloneqq \frac{\rho_i \rho_{i+1}}{\left( \sum_{j=2}^{n-1} \rho_j \rho_{j+1} \right)}.$$

Once $\tilde{\theta}_n$ is obtained, we use it to auto-regressively forecast the future performances. Particularly, we use $(\bar{J}_k)_{k=n+1}^{n+L}$ as the predicted performances for the next $L$ episodes, where

$$\forall i > n, \ \bar{J}_i \coloneqq f \left( (\bar{J}_{i-k}(\pi))_{k=1}^{p} ; \tilde{\theta}_n \right).$$

While our theoretical results were established for the setting where there is only a single regressor ($p = 1$), a more generalized theoretical result for $p > 1$ may be possible using the concepts of endogenous and exogenous regressors. Particularly, let $[..., X_i, X_{i+1}, X_{i+2}, X_{i+3}, ...]$, be observations from an $AR(2)$ time-series sequence where $X_{i+3}$ depends on $X_{i+1}$ and $X_{i+2}$. Here, using $X_{i+1}$ as the only instrument variable for $X_{i+2}$ is not possible as $X_{i+3}$ is correlated with $X_{i+1}$. However, $Z = X_i$ or even $Z = [X_i, X_{i+1}]$ may form a valid instrument for $X_{i+2}$ as neither the noise in $X_{i+3}$ nor the noise in $X_{i+2}$ is correlated with at least one component of $Z$, i.e., $X_i$. For precise instrument relevance conditions and additional discussion, we refer the reader to the works by Abbott [2007], Cameron [2019], Parker [2020]. We leave this theoretical extension for the future work.

### E.2 Implementation and Hyper-parameters

For the Pro-WLS baseline, we use the weighted least-squares procedure using the Fourier basis features [Chandak et al., 2020b]. The hyper-parameter for this baseline is the number of Fourier terms $d$ that should be used to estimate the performance trend. We found that setting $d$ to be too high results in extremely high-variance and setting it to a lower value fails to capture the trend in performance. Therefore, based on ablation studies in Figure 10 we set $d = 5$ for all the experiments.

WIS estimator uses all the data form the past. In comparison, for sliding windows WIS (SWIS), we set the sliding window length to be $400$ ($20\%$ of the number of episodes in the data) for all the experiments. That is, SWIS use past $400$ episodes to estimate the future performance.

For OPEN, the hyper-parameter corresponds to the number of terms to condition on during auto-regression. Similar to SWIS, we set $p = 400$ ($20\%$ of the number of episodes in the data) for all the experiments. That is the AR estimator uses past $400$ episodes to predict the performance in the

next episode. For the two stage regression, we observed that choice of learning rate, and avoiding over-fitting (using early-stopping) to be important as well.

For each environment, we collect data consisting of 2000 episodes of interaction using the behavior policy, and predict the expected future returns if executing the evaluation policy for the next 200 episodes. The behavior policy and the evaluation policy for each domain are described in Section E.3.

Since the future outcomes are stochastic, to evaluate the true expected future performance in (3), we create digital-clones of the environment *after* data has been collected using the behavior policy. Using these clones, we compute the average of 30 possible futures when executing the evaluation policy. This estimate of the *expected* future returns are then used as the ground truth for comparison with the predictions made by the algorithms.

For Figure 5, |bias| was computed using the absolute value of the difference between (a) the predicted future performance averaged across 30 trials and (b) the ground truth future performance. That is, for an estimator $\hat{J}$ of $J$, the bias is $|J - E[\hat{J}]|$. Because of this, 30 trials only gives us a point estimate for bias. (Notice that using the absolute value of the difference between (a) the predicted future performance for each trial and (b) the true future performance', averaged across 30 trials, will provide an estimate of $E[|J - \hat{J}|]$, which would not capture the bias but will be more like the variance (using L1/absolute distance instead of L2)).

### E.3 Environments

We provide empirical results on four non-stationary environments: a toy robot environment, non-stationary mountain car, diabetes treatment, and MEDEVAC domain for routing air ambulances. Details for each of these environments are provided in this section. For all of the above environments, we regulate the 'speed' of non-stationarity to characterize an algorithms' ability to adapt. Higher speed corresponds to a faster rate of non-stationarity; A speed of zero indicates that the environment is stationary.

**RoboToy:** This domain corresponds to the toy robot scenario depicted in Figure 1. Here, a robot can accomplish a task using either by 'running' or 'walking'. Robot finishes a task faster when 'running' than 'walking' and thus the reward received at the end of 'running' is higher. However, 'running' causes more wear and tear on the robot, thereby degrading the performance of both 'running' or 'walking' in the future. Since the past interactions influence the non-stationarity, this is an instance of active non-stationarity.

To perform more ablations on our algorithms, we also simulated a **RoboToy-Passive** domain, where there is no active non-stationarity as above. Instead, the reward obtained at the end of executing the options 'walking' or 'running' fluctuate across episodes. Therefore, the changes to the underlying system are independent of the actions taken by the agent in the past.

For both the active and passive version of this domain, we collect data using a behavior policy that chooses 'walking' more frequently, and the evaluation policy is designed such that it chooses 'running' more frequently.

**Non-stationary Mountain Car:** In real-world mechanical systems, motors undergo wear and tear over time based on how vigorously they have been used in the past. To simulate similar performance degradation, we adapt the classic (stationary) mountain car domain [Moore, 1990]. We modify the domain such that at every episode the effective acceleration force is decayed proportional to the average velocity of the car in the previous episode. This results in active non-stationarity as the change in the system is based on the actions taken by the agent in the past. Similar to the works by [Thomas, 2015, Jiang and Li, 2015], we make use of macro-actions to repeat an action 10 times, which helps in reducing the effective horizon length of each episode. The maximum number of step per episode using these macros is 30.

For our experiments, using an actor-critic algorithm [Sutton and Barto, 2018] we find a near-optimal policy $\pi$ on the stationary version of the mountain car domain, which we use as the evaluation policy. Let $\pi^{\texttt{rand}}$ be a random policy with uniform distribution over the actions. Then we define the behavior policy $\beta(o, a) := 0.5\pi(o, a) + 0.5\pi^{\texttt{rand}}(o, a)$ for all states and actions.

**Type-1 Diabetes Management:** Automated healthcare systems that aim to personalise for individual patients should account for the physiological changes of the patient over time. To simulate such a scenario we use an open-source implementation [Xie, 2019] of the U.S. Food and Drug Administration (FDA) approved Type-1 Diabetes Mellitus simulator (T1DMS) [Man et al., 2014] for the treatment of Type-1 diabetes, where we induced non-stationarity by oscillating the body parameters (e.g., rate of glucose absorption, insulin sensitivity, etc.) between two known configurations available in the simulator. This induces passive non-stationarity, that is, changes are not dependent on past actions.

Each step of an episode corresponds to a minute ($1440$ timesteps–one for each minute in a day) in an *in-silico* patient's body and state transitions are governed by a continuous time non-linear ordinary differential equation (ODE) [Man et al., 2014]. This makes the problem particularly challenging as it is unclear how the performance trends of policies vary in this domain when the physiological parameters of the patient are changed. Notice that as the parameters that are being oscillated are inputs to a non-linear ODE system, the exact trend of performance for any policy is unknown. This more closely reflects a real-world setting where Assumption 1 might not hold, as every policy's performance trend in real-world problems cannot be expected to follow *any* specific trend *exactly*–one can only hope to obtain a coarse approximation of the trend.

For our experiments, using an actor-critic algorithm [Sutton and Barto, 2018] we find a near-optimal policy $\pi$ on the stationary version of this domain, which we use as the evaluation policy. The policy learns the CR and CF parameters of a basal-bolus controller. Let $\pi^{\mathrm{rand}}$ be a random policy with uniform distribution over actions. Then we define the behavior policy $\beta(o, a) := 0.5\pi(o, a) + 0.5\pi^{\mathrm{rand}}(o, a)$ for all states and actions.

**MEDEVAC:** This domain stands for *med*ical *evac*uation using air ambulances. This domain was developed by Robbins et al. [2020] for optimally routing air ambulances to provide medical assistance in regions of conflict. This domain divides the region of conflict into $34$ mutually exclusive zones, and has $4$ air ambulances to serve all zones when an event occurs. Based on real-data, this domain simulates the arrival of different events, from different zones, where each event can have $3$ different priority levels. Serving higher priority events yields higher rewards. If an ambulance is assigned to an event, it will finish the assignment in a time dependent on the distance between the base of the ambulance and the zone of the corresponding event. While engaged in an assignment, that ambulance is no longer available to serve other events. A good controller decides whether to deploy, and which MEDEVAC to deploy, to serve any event (at the risk of not being able to serve a new high-priority event if all ambulances become occupied).

The original implementation of the domain assumes that the arrival rates of the events and the time taken by an ambulance to complete an event follow a Poisson process with a fixed rate. However, in reality, the arrival rates of different events can change based on external incidents during conflict. Similarly, the completion rate can also change based on how frequently an ambulance is deployed. To simulate such non-stationarity, we oscillate the arrival rate of the incoming high-priority events, which induces passive non-stationarity. Further, to induce wear and tear, we slowly decay the rate at which an ambulance can finish an assignment. This decay is proportional to how frequently the ambulance was used in the past. This induces active non-stationarity. The presence of both active and passive changes makes this domain subject to hybrid non-stationarity.

Similar to other domains, we used an actor-critic algorithm [Sutton and Barto, 2018] we find a near-optimal policy $\pi$ on the stationary version of this domain, which we use as the evaluation policy. Let $\pi^{\mathrm{rand}}$ be a random policy with uniform distribution over the actions. Then we define the behavior policy $\beta(o, a) := 0.5\pi(o, a) + 0.5\pi^{\mathrm{rand}}(o, a)$ for all states and actions.

### E.4   Additional Results

While the primary focus of this chapter was to develop methods to handle active/hybrid non-stationarity, we observed that the proposed method OPEN also provides benefits over the earlier algorithm Pro-WLS even when it is known that there is only passive non-stationarity in the environment.

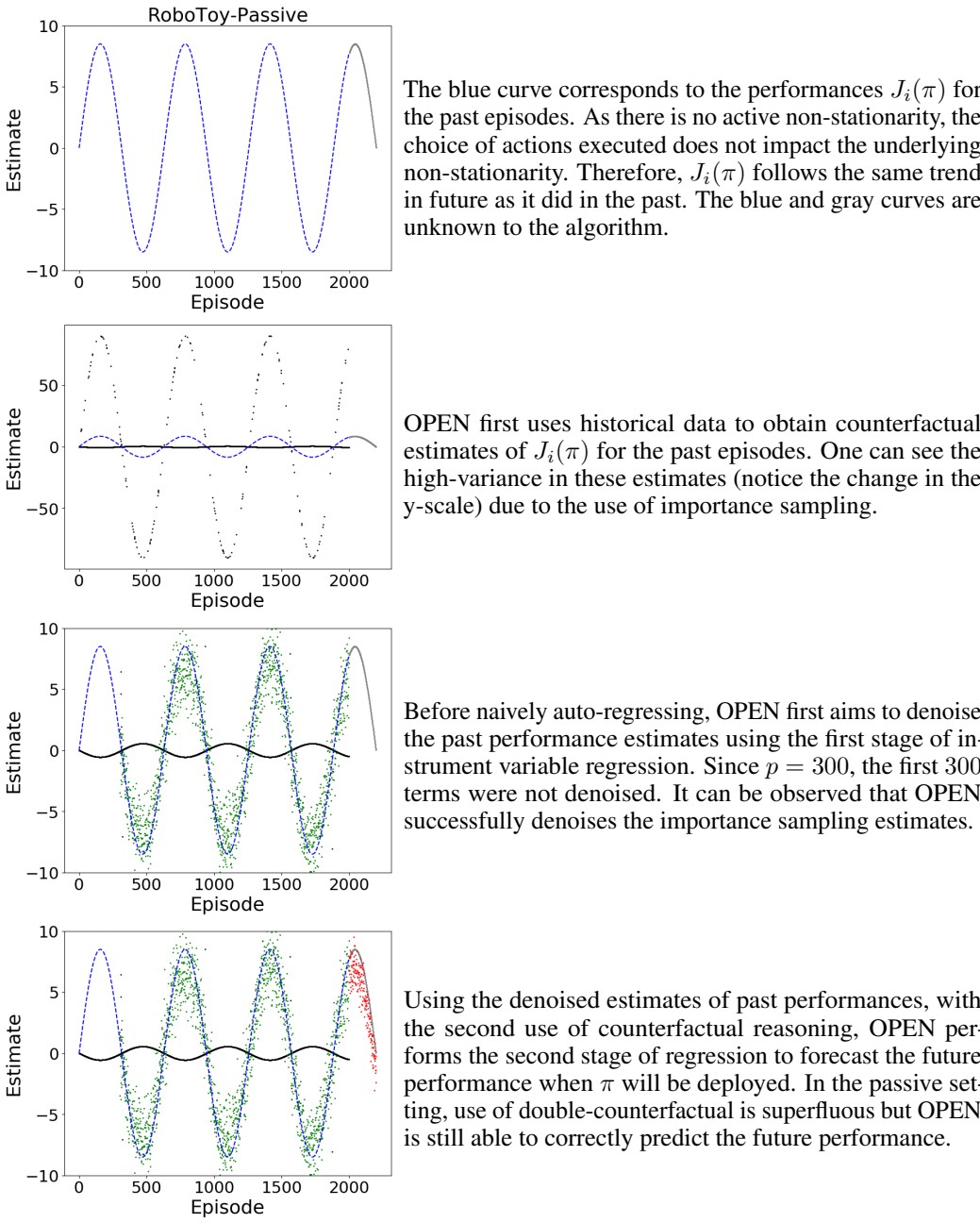

The blue curve corresponds to the performances $J_i(\pi)$ for the past episodes. As there is no active non-stationarity, the choice of actions executed does not impact the underlying non-stationarity. Therefore, $J_i(\pi)$ follows the same trend in future as it did in the past. The blue and gray curves are unknown to the algorithm.

OPEN first uses historical data to obtain counterfactual estimates of $J_i(\pi)$ for the past episodes. One can see the high-variance in these estimates (notice the change in the y-scale) due to the use of importance sampling.

Before naively auto-regressing, OPEN first aims to denoise the past performance estimates using the first stage of instrument variable regression. Since $p = 300$, the first 300 terms were not denoised. It can be observed that OPEN successfully denoises the importance sampling estimates.

Using the denoised estimates of past performances, with the second use of counterfactual reasoning, OPEN performs the second stage of regression to forecast the future performance when $\pi$ will be deployed. In the passive setting, use of double-counterfactual is superfluous but OPEN is still able to correctly predict the future performance.

Figure 8: An illustrative step by step breakdown of the stages in the proposed algorithm OPEN for the RoboToy-Passive domain.

### E.4.1 Single Run

Similar to Figure 4, in Figure 8 we present a step by step breakdown of the intermediate stages of a single run of OPEN on the RoboToy-Passive domain. Here the trend in how the performance of the evaluation policy was changing in the past remains the same in the future. When only passive non-stationarity is present, the double counter-factual correction performed by OPEN is superfluous. However, it can be observed that OPEN can still correctly identify the trend and provide useful predictions of $\pi$'s future performance.

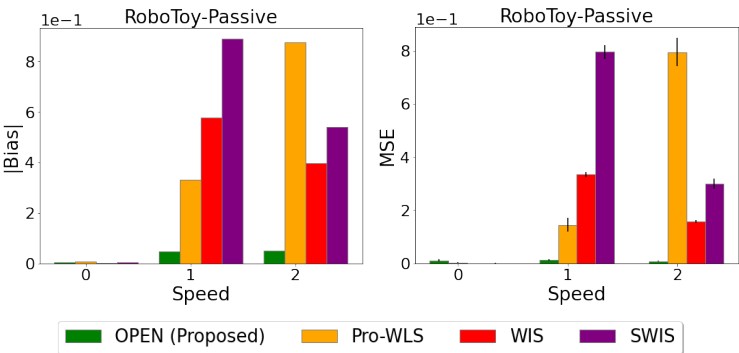

Figure 9: Comparison of different algorithms for predicting the future performance of evaluation policy $\pi$ on domains that exhibit passive non-stationarity. On the x-axis is the speed, which corresponds to the rate of non-stationarity; higher speed indicates a faster rate of change and a speed of zero indicates a stationary domain. **(TOP)** On the y-axis is the absolute bias in the performance estimate. **(Bottom)** On the y-axis is the mean squared error (MSE) in the performance estimate. **Lower is better** for all of these plots. For each domain, for each speed, for each algorithm, 30 trials were executed. Discussion of these plots can be found in Section E.4.

### E.4.2 Summary Plots

In Figure 9 we provide bias and MSE analysis of different algorithms on the domains that exhibit passive non-stationarity. Except for the stationary setting, where WIS has the best performance overall, we observe that for all other settings in the plot, OPEN performs better than both Pro-WLS and WIS consistently.

One thing that particularly stands out in these plots is the poor performance of Pro-WLS, despite being designed for the passive setting. We observed that because of the choice of parametric regression using the Fourier basis, Pro-WLS tends to suffer from high bias when the number of Fourier terms is not sufficient to model the underlying trend. Also, if the number of Fourier terms is increased naively, then they overfit the data and extrapolate poorly, thereby resulting in high-variance. In contrast, our method is based on an auto-regressive based time-series forecast that is more robust to the model choice (we kept the number of lag terms for auto-regression as $p = 300$ for OPEN for all our experiments).

To obtain all the results for Figure 5 and Figure 9, in total 30 different seeds were used for each speed of each domain for each algorithm to get the standard error. The authors had shared access to a computing cluster, consisting of 50 compute nodes with 28 cores each, which was used to run all the experiments.

### E.5 Ablation Study

In this section we study the sensitivity to hyper-parameters for the proposed method OPEN and the baseline method Pro-WLS [Chandak et al., 2020b]. The hyper-parameter for OPEN corresponds to the number of past terms to condition on for auto-regression, as discussed in Remark 3. The hyper-parameter for Pro-WLS corresponds to the order of Fourier bases required for parametric regression. In Figure 10 we present the results for how the performance of the methods vary for different choices of hyper-parameters.

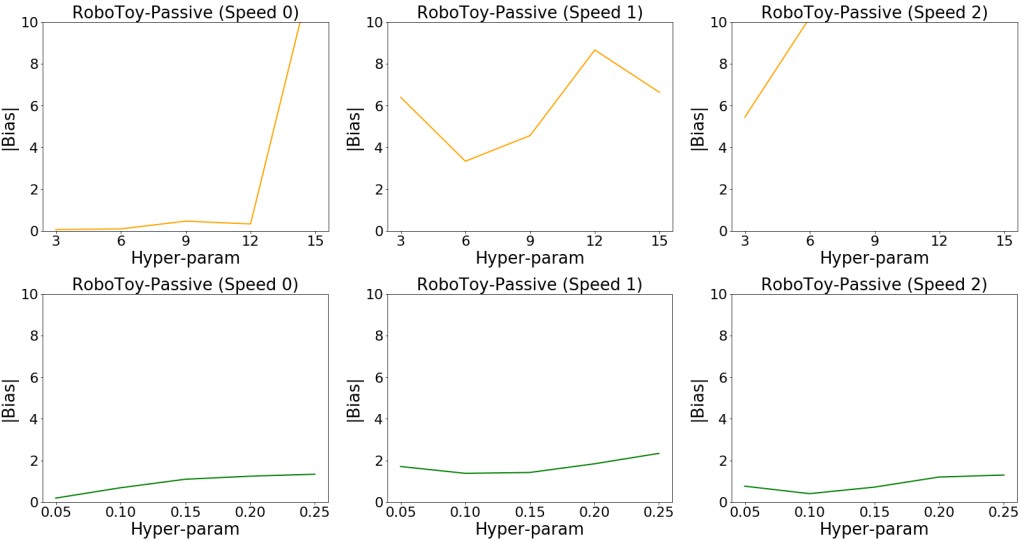

Figure 10: **(Top)** Absolute bias in prediction of Pro-WLS for different choices of its hyper-parameter. **(Bottom)** Absolute bias in prediction of OPEN for different choices of its hyper-parameter. For all the plots, lower value is better. Overall, we observe that OPEN being an auto-regressive method can extrapolate/forecast better and is thus more robust to hyper-parameters than Pro-WLS that uses Fourier bases for regression and is not as good for extrapolation.