# OpenReview forum: "Off-Policy Evaluation for Action-Dependent Non-stationary Environments"
_NeurIPS.cc/2022/Conference — NeurIPS 2022 Accept_

### Official Review · Reviewer_h3hM · 2022-07-04

**Rating:** 6
**Confidence:** 2
**Soundness:** 3 good
**Presentation:** 3 good
**Contribution:** 3 good

**Summary:**

This work proposes a method for evaluating (predicting) the future performance of a policy in non-stationary sequential decision processes. This allows to evaluated policies even if decision processes are changing over time. The work introduces an assumption on the structure of the changes in the decision process, which allows to bring forward statistical estimators for the off-policy performance even under changing decision processes. The benefit is demonstrated in four non-stationary environments.

**Questions:**

I still have some questions, in particular:
* In which timestep will the policy $\pi'$ that is introduced in Assumption 1 be followed? Is it the policy that was obeyed in POMDP $M_i$ or $M_{i+1}$? From the later exposition, I now think that is is the policy followed in $M_i$. Can the authors confirm this?
* It is not entirely clear how the bias calculated in Figure 5. I would appreciate if the authors could clarify this. If it is a stochastic variable, error bars could be added to the plot.


**Limitations:**

I think some of the limitations of this work are not sufficiently discussed. In particular, I wonder
* how sample efficient the method is
* how sensitive it is to violations of Assumption 1 and Assumption 2
* whether it can be used to learn optimal policies in non-stationary settings
I would have appreciated a discussion on some of these points. Societal impact is not discussed in the main paper; however, I also don’t see any negative implications of this work.


**Strengths And Weaknesses:**

## Strengths

* **Good Motivation.** I appreciate the motivation of this work with many examples such as motor war and past medical treatments that set the scene well. I agree that off-policy evaluation is an important problem in non-stationary decision processes.
* **Fair Assumptions.** As far as I can tell, Assumption 1 may be reasonable for many of the applications mentioned, but is still very general.
* **Theoretically grounded estimators.** The paper presents an estimator for the on off-policy performance in the future episode i+1, then removes its bias using instrumental variables regression and further introduces a technique to reduce the variance of this estimator. I think the overall techniques are valid in this context and result in a strong and consistent estimator.
* **Experiments are promising.** The authors present experiments on four artificial domains, that show significant improvement when deploying their method.

## Weaknesses
* **Related work section could be extended.** I think the related work section could be a little more extensive than the 5 works currently referenced. Even if this are the works most closely related to this particular paper, a reader not so familiar with sequential decision making /RL could benefit from some pointers to important works on reinforcement learning / MDPs / policy evaluation in general, for instance.
* **Notation was not always clear.** While the authors mostly did a good job with the formalization (admittedly, there are many notational objects), some notation is not entirely clear to me. For example, I don’t quite understand what $M_i(\pi)$ should be in Theorem 1 or what the expectation over $\beta_i, \beta_{i+1}$ should be interpreted in the same Theorem. I am not entirely familiar with the conventions in this subdomain, so this might not be a concern for a more proficient reader.
* **Comparison to single-sequence methods.** I agree that for some use-cases it is not possible to reset the entire environment and start a new sequence. However, for the RoboToy domain it would be possible to consider a life-long reward and reset it to the initial state several times. I would have found that comparison interesting in the experiment section. Even if there was some performance gap to the author's method, their approach still has the benefit that it can be applied to non-resettable environments.
* **The confounder for the IV regression may also affect** $J_{i-1}$. IV requires that the confounder is independent of the instrument. When using time-series data, this means that it affects $J_i$ and $J_{i+1}$, but not $J_{i-1}$. I wonder if this is a realistic assumption in this case. However, I admit that I have previously seen this assumption being used in time series modelling, but maybe the authors can elaborate on why they think it’s plausible in this particular case.

## Minor
* In the definition below l. 203 (it would be great if all equations could be numbered), I am uncertain if really an expectation via the policies $\pi'$ is taken. I don’t see how this definition should be understood, however I agree with the right-hand side.
* The authors state assumption 2 is “standard” but do not provide any references. I think adding some references that make the same assumption could strengthen the argumentation.

## Disclaimer
The reviewer admits that despite their best efforts, their knowledge in the domain of sequential decision making / RL is limited. Therefore they were not able to assess in particular: The novelty of the method, the significance of the problem for the domain, the empirical strength of the experiments.

---

> ### Author Response · Authors · 2022-08-02
> **Response to Reviewer h3hM (1/2)**
>
> Thank you for your great effort in reviewing our paper and providing insightful comments. We are glad that you recognized the importance of dealing with non-stationarity for real-world off-policy evaluation, and found the assumptions reasonable. Your comments have made us improve the paper and make it more self-contained. In the following we address your comments one by one.
>
> 1. **“Related work section could be extended.”**
>
> Thank you for the suggestion. We have extended the related work both in the main body and in the Appendix B to provide additional pointers on important works on reinforcement learning and off-policy evaluation.
>
>
> 2. **“Comparison to single-sequence methods.”**
>
> That’s an interesting thought. If we reset the environment to generate multiple independent trajectories and use existing methods (for e.g,, ordinary/weighted importance sampling) to estimate a policy’s performance then it would provide an estimate of that policy’s long-term performance conditioned on that policy being executed from the start. However, the objective of interest is $\mathscr J(\pi)$ that is defined as $\pi’s$ future performance conditioned on the interactions so far (as defined in the problem statement). It is not immediate how to leverage the resetting procedure for estimating $\mathscr J(\pi)$ and compare against other methods that are also estimating $\mathscr J(\pi)$ but without resetting. We are not yet clear exactly what the proposed continuous method would be that the reviewer is thinking of– **could you please expand a bit on your suggestion?**
>
>
> 3. **“The confounder for the IV regression may also affect J_{i-1}“**
>
> Indeed, one use of IV is when the confounder is independent of the instrument. Or put differently, the noise in the $X$ variable (due to confounder) is independent of the instrument. However, (a) the independence condition can be relaxed to only require the instrument to be uncorrelated with the noise in the $X$ variable, and (b) In our case, $X$ is the current performance and noise in $X$ results due to availability of only unbiased estimates of $X$ due to importance sampling. As this noise is not due to any confounding but due to importance sampling error, we do not have to worry about the confounder. Perhaps another way to think about this setup might be from the point of view of IVs for errors-in-variable model [https://en.wikipedia.org/wiki/Errors-in-variables_models#Simple_linear_model](https://en.wikipedia.org/wiki/Errors-in-variables_models#Simple_linear_model). Further, Theorem 2 establishes that instrument and X are uncorrelated in our setup.
>
> 4. **“ state assumption 2 is “standard” but do not provide any references”**
>
> Thank you for pointing this out. We have included the reference for existing works on off-policy evaluation in the stationary setting that make this assumption [1,2,3,4].
>
> [1] Thomas, Philip, Georgios Theocharous, and Mohammad Ghavamzadeh. "High-confidence off-policy evaluation." Proceedings of the AAAI Conference on Artificial Intelligence. Vol. 29. No. 1. 2015.
>
> [2] Thomas, Philip, and Emma Brunskill. "Data-efficient off-policy policy evaluation for reinforcement learning." International Conference on Machine Learning. PMLR, 2016.
>
> [3] Xie, Tengyang, Yifei Ma, and Yu-Xiang Wang. "Towards optimal off-policy evaluation for reinforcement learning with marginalized importance sampling." Advances in Neural Information Processing Systems 32 (2019).
>
> [4] Kallus, Nathan, and Masatoshi Uehara. "Double reinforcement learning for efficient off-policy evaluation in markov decision processes." Journal of Machine Learning Research 21.167 (2020).
>
> 5. **“It is not entirely clear how the bias calculated in Figure 5.”**
> Thank you for highlighting this was unclear: We have update Appendix E.2 to mention that |bias| was computed using the absolute value of the difference between  (a) the predicted future performance averaged across 30 trials and (b) the true future performance. That is, for an estimator $\hat J$ of $J$, the bias is  $|J - E[\hat J]|$. Because of this, 30 trials only gives us a point estimate for bias.
>
> (Notice that using the absolute value of the difference between  (a) the predicted future performance for each trial and (b) the true future performance’, averaged across 30 trials, will provide an estimate of $E[|J - \hat J|]$, which would not capture the bias but will be more like the variance (using L1/absolute distance instead of L2))
>
> 6. **“how sample efficient the method is”**
>
> As the current work takes the first steps towards an OPE method to tackle both active and passive non-stationarity, we focused on developing the main algorithm, investigating bias-variance trade-off and analyzing asymptotic consistency. Developing finite sample guarantees would be an interesting future work.

---

> ### Author Response · Authors · 2022-08-02
> **Response to Reviewer h3hM (2/2)**
>
>
> 7. **“how sensitive it is to violations of Assumption 1 and Assumption 2”**
>
> We will emphasize in the main body that Appendix C provides additional plots and discussions regarding the (in)validity of Assumption 1. Further, we will also refer the readers to the work by Thomas et al. [5] that discusses settings where Assumption 2 can be relaxed.
>
> [5] Thomas, Philip S., and Emma Brunskill. "Importance sampling with unequal support." Thirty-first AAAI conference on artificial intelligence. 2017.
>
>
> 8. **“whether it can be used to learn optimal policies in non-stationary settings I would have appreciated a discussion on some of these points.”**
>
> That’s a really interesting point. We think it should be possible to do policy improvement by computing the gradient of the predicted future performance with respect to the parameters of the evaluation policy, similar to Section 5.2 in the work by Chandak et al. [6]. However, since our proposed method uses importance-weighted IV based techniques to tackle settings with both active and passive non-stationarity, the exact steps for computing the required gradient is not immediate. We plan to investigate this in the future.
>
>
> [6] Chandak, Yash, et al. "Optimizing for the future in non-stationary mdps." International Conference on Machine Learning. PMLR, 2020.
>
>
>
> **Notation:**
>
> 9. **“E_{beta_i,beta_i+1}“**
>
> We should have and will add more clarification on this notation. This notation was used to indicate that interactions with the environment in the $i^\text{th}$ episode was done using policy $\beta_i$. At the end of this interaction, the environment changes from $M_i$ to $M_{i+1}$. Subsequently, policy $\beta_{i+1}$ is used to interact with $M_{i+1}$.
>
> 10. **What does M_i(pi) mean exactly?**
>
> Thank you for catching this. This was a typo–it should just be $M_i$ (the environment corresponding to the $i^{th}$ episode).
>
> 11. **“I am uncertain if really an expectation via the policies $\pi’$ is taken”**
>
> Thank you for pointing out that this definition can be clarified. We will update the draft to mention that the meta-transition function ($\mathcal T$ defined under the notation section) that governs the change in the environment is dependent on the interactions by the policy. Therefore, the expectation in this definition corresponds to the expected performance in the next episode across the possible next environments induced due to stochastic interactions by the policy $\pi’$ in the current environment.
>
> 12. **“In which timestep will the policy that is introduced in Assumption 1 be followed?..., I now think that is is the policy followed in $M_i$”**
>
> Yes, that’s correct. We will add a sentence below the assumption to make this clear for future readers.

---

### Official Review · Reviewer_3Ye2 · 2022-07-10

**Rating:** 4
**Confidence:** 3
**Soundness:** 3 good
**Presentation:** 2 fair
**Contribution:** 2 fair

**Summary:**

This paper proposes an off-policy evaluation method for non-stationary environments. The authors distinguish between active and passive non-stationarity, and use double counterfactual reasoning and importance-weighted instrument-variable regression to obtain bias and variance estimation. Empirical comparisons show the superiority of the proposed OPEN method to other baselines in terms of predict future performances.

**Questions:**

1.	I’m confused on the definition of “active non-stationarity”. For example, in the discussion of automated healthcare in Introduction, public health continuously evolving based on the treatments made available in the past. Wouldn’t be just state transitions as discussed in MDP? In an MDP, state changes and yet the transition function can stay stationary. In a similar manner, even the title “Action-Dependent Non-Stationary” can be confusing, as my interpretation of this is just state transition based on actions. Can the author please clarify what changes over time and what is non-stationary in a more rigorous mathematical definition?

2.	Similarly, the authors claim that this is the first work towards off-evaluation for non-stationary environments and thus also in the empirical analysis, only baselines for passive non-stationarity or no-stationarity are used. However, to me, the active non-stationarity considered in this work is similar to that in reinforcement learning. Can the authors comment on the difference/contribution of your work and, for example, [1,2,3,4] for off-policy evaluation for reinforcement learning? I would also suggest to use these works as more stronger baselines for empirical comparison.


Reference
[1] Xie, T., Ma, Y. and Wang, Y.X., 2019. Towards optimal off-policy evaluation for reinforcement learning with marginalized importance sampling. Advances in Neural Information Processing Systems, 32.
[2] Thomas, P. and Brunskill, E., 2016, June. Data-efficient off-policy policy evaluation for reinforcement learning. In International Conference on Machine Learning (pp. 2139-2148). PMLR.
[3] Munos, R., Stepleton, T., Harutyunyan, A. and Bellemare, M., 2016. Safe and efficient off-policy reinforcement learning. Advances in neural information processing systems, 29.
[4] Jiang, N. and Li, L., 2016, June. Doubly robust off-policy value evaluation for reinforcement learning. In International Conference on Machine Learning (pp. 652-661). PMLR.


**Limitations:**

There is no discussion of limitation. Along with my questions regarding the difference between this work and reinforcement learning, one possible direction in the conclusion is to talk about the similarity and difference, and to what extent the results are generalizable to RL settings.

**Strengths And Weaknesses:**

Strengths:
1.	The structure of this paper is clear. This paper addresses an interesting question of off-policy evaluation under the presence of non-stationarity.
2.	The presented algorithm is well grounded from Econ theories by utilizing instrument variables.

Weakness:
1.	Literature review is not adequate. Even with the content in the appendix, this is no discussion of off-policy evaluation for reinforcement learning or non-stationary multi-armed bandits.
2.	The claim and the discussion of “active non-stationarity” is somewhat confusing (more on this later in Questions).
3.	The authors seem to overstate their contribution.
4.	Baseline methods are weak and not presenting state-of-the-art.

---

> ### Author Response · Authors · 2022-08-02
> **Response to Reviewer 3Ye2**
>
> Thank you for your great effort in reviewing our paper and for your helpful comments. Your comments have helped us understand and clarify possible points of confusion a reader might have. We are glad to hear that you found the proposed method well grounded. In the following we address your comments one by one.
>
> 1. **“ Even with the content in the appendix, this is no discussion of off-policy evaluation for reinforcement learning or non-stationary multi-armed bandits.”**
>
> Thank you for pointing this out. Since our work looks at the problem of non-stationarity in the sequential decision making setup, we had focused related work along that direction. Nonetheless, we agree that some readers might also benefit from additional discussion and references on prior work related to off-policy evaluation in the stationary setting, or non-stationarity in the bandit/single-decision setting. We have updated both the main body and the Appendix B to include a discussion of the works you have referenced and other similar works.
>
>
> 2. **Response to Q1 and Q2:**
>
> Indeed, like you mentioned, often the first instinct might be to consider the problem as a stationary environment with a single long sequence of interactions. However, the key challenge is that predominantly the existing off-policy evaluation methods (including [1,2,3,4]) assume availability of either (a) resetting assumption to sample multiple sequences of interactions from a stationary environment with a fixed starting state distribution (i.e., episodic setting), or (b) ergodicity assumption such that interactions can be sampled from a steady-state/stationary distribution (i.e., continuing setting). For the problems of our interest, methods based on these assumptions may not be viable. For e.g., in automated healthcare, we have a single long history for the evolution of public health, which is neither in a steady state distribution nor can we reset and go back in time to sample another history of interactions. Thank you for pointing out this potential source of confusion, we have updated the introduction to bring the above points to a reader’s attention.
>
> Because of the above reasons we model the problem using a non-stationary setup where the agent interacts with a sequence of POMDPs, where the changes are governed by a meta transition function $\mathcal T$ that is formally discussed in the notations.
>
> Based on your comments, we have also updated Appendix B.1 and B.4 to include additional related work. Particularly, Xie et al. [1] and Munos et al. [3] require full-observability of the state and are thus not applicable to our setting where we only have partial observability. Similarly, work by Thomas et al. [2] and Jiang et al. [4] show how in the stationary setting partial models of the environment can be combined with importance sampling based estimators. As in our work we do not assume any access to the model of the environment, we directly compare against the importance sampling based estimators. Furthermore, a thorough review about the non-stationary setting is also available in the survey papers  by Padakandla [5], Khetarpal et al. [6].
>
> **Please do let us know if we could further clarify any differences between the proposed work and the conventional off-policy evaluation methods in reinforcement learning.**
>
>
> [1] Xie, T., Ma, Y. and Wang, Y.X., 2019. Towards optimal off-policy evaluation for reinforcement learning with marginalized importance sampling. Advances in Neural Information Processing Systems, 32.
>
> [2] Thomas, P. and Brunskill, E., 2016, June. Data-efficient off-policy policy evaluation for reinforcement learning. In International Conference on Machine Learning (pp. 2139-2148). PMLR.
>
> [3] Munos, R., Stepleton, T., Harutyunyan, A. and Bellemare, M., 2016. Safe and efficient off-policy reinforcement learning. Advances in neural information processing systems, 29.
>
> [4] Jiang, N. and Li, L., 2016, June. Doubly robust off-policy value evaluation for reinforcement learning. In International Conference on Machine Learning (pp. 652-661).
>
> [5]  Padakandla, Sindhu. "A survey of reinforcement learning algorithms for dynamically varying environments." ACM Computing Surveys (CSUR) 54.6 (2021): 1-25.
>
> [6] Khetarpal, Khimya, et al. "Towards continual reinforcement learning: A review and perspectives." arXiv preprint arXiv:2012.13490 (2020).
>
> 3. **“There is no discussion of limitation.”**
>
> We will emphasize in the main body that due to space constraints we had deferred discussion of limitations and potential negative impacts to Appendix A.6.
>
> 4. **“similarity and difference, and to what extent the results are generalizable to RL settings.”**
>
> Thank you for this suggestion. We will emphasize under the notations that our considered setup directly generalizes off-policy evaluation problems in RL without requiring either the resetting or the ergodicity assumption.

---

> > ### Comment · Reviewer_3Ye2 · 2022-08-10
> > **Thanks for the rebuttal**
> >
> > I appreciate the response from the authors. While most of my questions/concerns are addressed, I'd like to see more explicit discussion on Q1 regarding the definition of "active non-stationarity” which seems lacking from the paper/rebuttal. I'm worried about potential overlaps with existing work in MDP.

---

> ### Author Response · Authors · 2022-08-08
> **Response to Reviewer 3Ye2**
>
> We thank the reviewer for their time to provide thoughtful suggestions and questions. As the author's rebuttal window is about to close, we just wanted to ensure that our previous response was able to address the questions. We would be happy to answer if there are any more questions. Thank you!

---

### Official Review · Reviewer_Gp7U · 2022-07-11

**Rating:** 5
**Confidence:** 4
**Soundness:** 3 good
**Presentation:** 3 good
**Contribution:** 2 fair

**Summary:**

The paper proposes an off-policy evaluation algorithm for non-stationary environments. Compared to prior work, where non-stationarity is assumed to be external to the agent's actions, this work proposes a setup where the non-stationarity can also come from the agent's own actions. Under a few assumptions, the paper derives a new algorithm which combines importance weighting and IV regression. The paper details a number of theoretical results regarding the consistency of the resulting estimate and shows the new estimate outperforms baseline methods which do not account for the non-stationarity resulting from actions.

**Questions:**

=== Assumption 1

The presentation of Assumption 1 is a little bit confusing. Does Eqn 1 hold for all pi' or a particular pi'? Given a fixed pi_1, pi_2, if Eqn 1 holds for both pi_1 and pi_2 we define the corresponding equality to be L_1, L_2, should we expect L_1=L_2 as well? From the fact that Assumption 1 implies the passive non-stationarity in line 169, it seems that we should also have equality between different pi's?

The notation P(J=J_{i+1}|M=m;pi') is also a bit confusing at first glance and might need some more explanations. Concretely, what does pi' mean in the notation? I presume that the notation's dependency on pi' means that we execute pi' at the i-th episode, and end in (i+1)-th episode -- in other words, the change from episode i to episode i+1 is due to the execution of policy pi' instead of pi. Is this correct?

=== Theorem 1

The notation E_{beta_i,beta_i+1} is a bit confusing at first glance. Does it mean that we execute beta_i when in M_i, transition to a POMDP M_{i+1} and then execute beta_{i+1} on top of M_{i+1}? It is worth making the notation more clear and maybe spelling out the explicit interactions here.

What does M_i(pi) mean exactly? Does it refer to the performance one obtains by executing pi in POMDP M_i? It is not immediately clear to me what's the difference between M_i(pi) and J_i(pi), are they both random variables? The two quantities on both sides of Theorem 1, are they random variables or fixed scalars?

=== Theorem 2 and Assumption 3

Thm 2 is indeed quite a surprising result. It is better to offer some more explanations behind the intuition and result of Thm 2. The proof in Appendix D line, it is not immediately clear to me why E_beta[hat{J}_{i+1}|hat{J}_i] = J_{i+1}.

It also seems that the proof for Thm 2 does not require Assumption 2? It is more clear if during the proof when Assumption 1/2 are applied they are explicitly cited.

=== IV regression for bias reduction

On a high level, Eqn 3-4 suggests that instead of directly carrying out regression X->Y, we should do something like

X->g
g->Y

The two stage process accumulates X into an IV g, and then use g to regress against Y to avoid bias. Is there more familiar example in RL which uses this kind of two time-scale like approximation? It might be helpful to spell out explicitly that here g is the IV. Is the intuition here essentially that by regressing X->g, we allow g to capture the structured information in X while decorrelates itself with random noise in X (in the paper's notation, decorrelate with input noise eta)

=== Empirical analysis in Fig 4

Fig 4 is quite difficult to parse even after reading the texts below the plots.

On the right plot, is it the case that here green dots are essentially values produced by the function g? Is there a reason why the green dots still have quite some variations for any given episode, since I'd expect a smoother prediction function g.

On the right plot, I do not see black dots in the right plot and can only see green dots, which correspond to the prediction by the IV regression.

On the middle plot, is the black curve meant to show that IS weighted return generally has very high variance? The bottom curve I cannot see clearly, maybe better to use log scale.

=== Fig 5

It is good that accounting for active stationarity reduces bias in general. However, in MEDEVAC plot on the bottom, when speed=2, Pro-WLS performs significantly better than OPEN in terms of MSE. Is this because OPEN is hurt by variance? All three methods should all arguably apply certain levels of importance sampling or normalized IS, why should OPEN be more sensitive to variance in this case? It might be useful to spell out an explicit algorithmic difference between Pro-WLS and OPEN to better see the distinction (juxtapose two algo boxes).

Better use log scale as some bars are barely visible.

=== POMDP method comparison

The paper has discussed in a few places that a naive POMDP approach to modeling stationarity is not feasible, because the visitation assumption is violated. I think it'd be useful to run a naive POMDP algorithm and see how it behaves on these tasks, to see if violation of the theoretical assumption really hurts the performance in practice.

**Limitations:**

The paper has discussed a number of its limitations.

**Strengths And Weaknesses:**

=== Strengths

The paper takes a first step towards addressing non-stationarity due to the agent's own interactions, and hence takes a relatively significant conceptual step forward compared to prior work. The paper is largely clear in its presentation, has fairly strong theoretical and empirical results that showcase the new algorithm's advantage over baselines.

=== Weaknesses

I think maybe the major weakness is that Assumption 1 feels quite strong, which on a high level, requires that the meta-transition matrix of the agent's performance looks like a MDP transition matrix. Though the assumption does capture the fact that "the way the changes happen is fixed", it seems to be quite restrictive in an intuitive sense. I would appreciate more discussion on the limitation of this assumption and how maybe some applications do not fit into such an assumption.

---

> ### Author Response · Authors · 2022-08-02
> **Response to Reviewer Gp7U (1/2)**
>
> Thank you for your great effort in reviewing our paper and for your precise questions. We are glad that you recognized the importance of dealing with non-stationarity for real-world off-policy evaluation, and found our work conceptually significant. Your comments have made us significantly improve the paper for future readers. In the following we address your comments one by one.
>
> 1. **“...I would appreciate more discussion on the limitation of this assumption [1] and how maybe some applications do not fit into such an assumption.”**
>
> Thank you for the good suggestion. We agree it is (almost always!) preferable to have less restrictive assumptions but as we discuss around Line 50, without any assumptions it may be impossible to make any reliable estimates in this setting. Therefore, we focused on exploiting structure on how the performance of a policy changes over time. As we discuss in Remark 1: Assumption 1 can be relaxed to condition on multiple past performances thereby allowing to capture a wide variety of possible trends. For instance, Figure 6 in Appendix C presents different performance trends that can be approximated, illustrating that the assumption is quite general. However, as the reviewer notes, the assumption is not completely flexible. For instance, Figure 6 (right) discusses a setting where this assumption does not hold: Consider a motor of an industrial system that is degrading over time but this degradation has no effect on the observable performance, until the point when the motor breaks down and the performance drops completely. In such cases, just looking at past performances may not be sufficient to infer how performance will change in the future.
>
>
> Due to space constraints, this additional discussion on the assumption and the cases where this assumption is (in)valid was deferred to Appendix C. We will emphasize Remark 1 to highlight this.
>
> 2. **“ Does Eqn 1 hold for all pi' or a particular pi'?…”**
>
> Thank you for noticing this. We should have and will clarify in the draft that for all the theoretical results, Eqn 1 only needs to hold for the $\pi’$ corresponding to the policy whose future performance needs to be evaluated.
>
> 3. **“Given a fixed pi_1, pi_2, if Eqn 1 holds for both pi_1 and pi_2 we define the corresponding equality to be L_1, L_2, should we expect L_1=L_2 as well?“From the fact that Assumption 1 implies the passive non-stationarity in line 169, it seems that we should also have equality between different pi's?”**
>
> We apologize but we are not sure if we understood the question properly. Actually Assumption 1 does not necessarily imply passive non-stationarity; In the current setup, Assumption 1 can _also_ model passive non-stationarity as for that setting conditioning on $\pi’$ is superfluous. To see this, consider an example in the stationary MDP setting: Pr(S_{t+1}|S_t=s, A_t=a, S_{t-1}=x) = Pr(S_{t+1}|S_t=s, A_t=a, S_{t-1}=y). Here conditioning on $S_{t-1}$ is superfluous and this equality will not imply equality between $x$ and $y$ either.  **Does this address your question?**
>
>
> 4. **“It is better to offer some more explanations behind the intuition and result of Thm 2…”, “ Thm 2 does not require Assumption 2”**
>
> Thank you for pointing out that this result can benefit from more clarification. We have elaborated the proof steps in the appendix and mentioned where Assumption 2 is used. Intuitively, $H_i$ and $H_{i+1}$ are dependent because $H_i$ can influence which environment $M_{i+1}$ the agent will see in episode $i+1$. However, given an environment, if the data collection is done using a policy $\beta$ that satisfies Assumption 2, then the importance sampling based estimator provides unbiased estimates for the performance of $\pi$ in that environment**. Therefore the noise term $\widehat J_{i+1}(\pi) - J_{i+1}(\pi)$ is always mean zero irrespective of how the agent came to $M_{i+1}$. Because of this the covariance between the past performance estimate and the above noise term is always zero.
>
> ** Detailed steps for this can be found in Eqn 3.4 and 3.5 in the work by Thomas [1].
>
> [1] Thomas, Philip S. "Safe reinforcement learning." (2015).

---

> ### Author Response · Authors · 2022-08-02
> **Response to Reviewer Gp7U (2/2)**
>
> 5. **“ IV regression for bias reduction”**
>
> Perhaps one familiar work in RL that uses IV is by Bradtke and Barto [1]. However, their usage of IV is for analysis of TD methods and is quite different from ours.
>
> For intuition, indeed, the high-level idea is that instead of regressing from X -> Y (where X is the current performance and Y is the future performance),  we can use an instrument variable Z (in our case it is the estimate of performance in the past) to denoise X before performing the above regression.
>
> [1] Bradtke, Steven J., and Andrew G. Barto. "Linear least-squares algorithms for temporal difference learning." Machine learning 22.1 (1996): 33-57.
>
>
> 6. **“... Is there a reason why the green dots still have quite some variations for any given episode….”, “On the right plot, I do not see black dots….”, “On the middle plot,...”**
>
> Thank you for the feedback on the plots. We have emphasized in the caption that the scale of the Y-axis is different for the middle plot. The middle plot is indeed aimed at showing the high-variance of the IS weighted return (black dots). The right-plot uses a different scale for the Y-axis to focus on the estimates obtained (green dots) using the proposed IV technique. Because of this all the black dots have values outside the limits of the Y-axis on the right plot. In comparison, the spread of green dots in the right figure is almost **2 orders of magnitude less** than the spread of black dots in the middle plot.
>
> 7. **“... Is this because OPEN is hurt by variance? “**
>
> Indeed, even though all three methods apply normalized IS in different ways, in this specific case OPEN procedure incurs higher variance. The instrument variable method helps in de-biasing (as can be seen from the bias plots for MEDEVAC) but may increase the variance of the estimator. This can be explained by observing the closed form equation in Eqn 22 obtained using the IV procedure. Here, $Z$ is the instrument variable and if it is weakly correlated with X (i.e,. $Z^\top X$ has a small magnitude) then $(Z^\top X)^{-1}$ can be large thereby increasing variance. We have updated Appendix A.6 to reflect this.
>
> 8. **“... a naive POMDP algorithm”**
>
> In this work we focused on both theoretically and empirically studying methods for the passive and active non-stationary setting that do not require inferring the unobserved state/latent-variable at all. Hence we compared with algorithms that are the most similar. In Appendix C.1 we provide discussion for the latent variable setting and how it may not even be possible to infer the latent variables in the completely offline setting like ours (unless more assumptions are imposed). For future work we plan to study in more detail such methods that require inferring latent variables.
>
>
> **Notations:**
>
> 9. **“... P(J=J_{i+1}|M=m;pi') …” and “E_{beta_i,beta_i+1}“**
>
> Your interpretation is accurate. We will update the notation section to add more clarification on these. Further, as you suggested, we will also spell out the explicit interactions when we use it.
>
> 10. **"What does M_i(pi) mean exactly?"**
>
> Thank you for catching this typo. It should just be $M_i$, the environment in the $i^th$ episode. We have fixed this.
>
>
> 11. **"The two quantities on both sides of Theorem 1, are they random variables or fixed scalars?"**
>
> They are fixed scalars. Thanks for pointing out the potential confusion and we have updated the sentence of Theorem 1 to make this clear.

---

> ### Author Response · Authors · 2022-08-08
> **Response to Reviewer Gp7U**
>
> We thank the reviewer for their insightful and precise questions. As the author's rebuttal window is about to close, we just wanted to ensure that our previous response was able to address the questions. We would be happy to answer if there are any more questions. Thank you!

---

### Official Review · Reviewer_crSm · 2022-07-11

**Rating:** 7
**Confidence:** 3
**Soundness:** 3 good
**Presentation:** 3 good
**Contribution:** 3 good

**Summary:**

This paper studies the problem of policy evaluation when the environment is changing over episodes. The three ideas that are used to construct an accurate estimator is (1) importance sampling, (2) bias reduction, and (3) variance reduction. Under some reasonable assumptions, the authors show strongly consistency of the constructed estimator. Numerical experiments demonstrate that the proposed algorithm (called OPEN) performs well in several artificial environments.

**Questions:**

Please see the previous box.

**Limitations:**

This paper does not have any negative potential negative societal impact.

**Strengths And Weaknesses:**

Overall I think this paper is novel. The setting where the environment may change over episodes is a reasonable extension of standard policy evaluation. There can be many other interesting motivating examples. In autonomous driving, the performance of the same policy on the same car may be different depending on how long the car is used.

(1) Suppose we construct a new state-space as $S\times M$, i.e., we view $M_i$ as an additional state. By doing this do we recover the standard policy evaluation setup where the environment is stationary?

(2) Assumption 1 is somewhat restrictive. It says that the change of the environment depends only on the performance of the previous policy. What is the main challenge in the analysis without this assumption?

(3) There are many related work studying importance sampling based off-policy TD-learning type of algorithms, such as [1,2,3,4], where various techniques are used to overcome the high variance issue in off-policy sampling.

[1] Harutyunyan, A., Bellemare, M. G., Stepleton, T., & Munos, R. (2016, October). Q ($\lambda $) with Off-Policy Corrections. In International Conference on Algorithmic Learning Theory (pp. 305-320). Springer, Cham.

[2] Precup, D. (2000). Eligibility traces for off-policy policy evaluation. Computer Science Department Faculty Publication Series, 80.

[3] Munos, R., Stepleton, T., Harutyunyan, A., & Bellemare, M. (2016). Safe and efficient off-policy reinforcement learning. Advances in neural information processing systems, 29.

[4] Espeholt, L., Soyer, H., Munos, R., Simonyan, K., Mnih, V., Ward, T., ... & Kavukcuoglu, K. (2018, July). Impala: Scalable distributed deep-rl with importance weighted actor-learner architectures. In International conference on machine learning (pp. 1407-1416). PMLR.

---

> ### Author Response · Authors · 2022-08-02
> **Response to Reviewer crSm**
>
> Thank you for your great effort in reviewing our paper and providing insightful comments. We are glad that you recognized the importance of dealing with non-stationarity for real-world off-policy evaluation, and found our work novel. Your comments have helped us make the work more self-contained. In the following we address your comments one by one.
>
>
> 1. **“Suppose we construct a new state-space as $S\times M$  i.e., we view $M_i$  as an additional state…”**
>
> Assuming that $M_i$ refers to a POMDP (using the notation from the paper), indeed if one could embed the entire POMDP in a given state then it may make this resulting ‘new’ environment stationary. Although additional care must be taken for the policy evaluation problem as a policy’s performance is defined using the starting state distribution $\mu_i$ (defined under notations). In general, this distribution can keep changing as well and can thus re-introduce non-stationarity for the policy evaluation problem even for the ‘new’ environment considered above.
>
> We will also update the main draft to refer readers to Appendix C.1, which contains discussion on a related hypothetical setting where a ‘new’ environment is constructed using some latent variable $z_i$ associated with $M_i$.
>
> 2. **“...  It says that the change of the environment depends only on the performance of the previous policy…?”**
>
> That’s a great question. We will emphasize the discussion of $\mathcal T$ (under Notation section) to better highlight that the change of the environment can actually depend on both (a) the previous environment and (b)  the entire sequence of interactions in that previous environment. Intuitively, Assumption 1 only says that the past performance provides enough information to model the distribution of the current performance. In terms of analysis, the double counterfactual reasoning process to obtain unbiased estimates of a policy’s future performance given its previous performance may not have been possible without this assumption.
>
> 3. **“... various techniques are used to overcome the high variance issue in off-policy sampling.”**
>
> Thank you for sharing these works that discuss ways to tackle high variance in the stationary setting. We have included these in the related work in the main body and also extended the related work in Appendix B.

---

> > ### Comment · Reviewer_crSm · 2022-08-09
> > **Response to Authors**
> >
> > Thank the authors for their response. I do not have further questions.

---

### Meta-Review · Area_Chair_ytW2 · 2022-08-26

**Recommendation:** Accept
**Confidence:** Certain

**Metareview:**

The paper addresses off-policy evaluation for non-stationary environments. A key distinction from the prior work, highlighted by multiple reviewers is the aspect that non-stationarity can come from the agent’s own actions (as opposed to just the exogenous factors). The paper uses a combination of importance sampling and bias and variance reduction to show strong consistency of the derived estimator.

There are certain assumptions that were considered too restrictive by a reviewer (re: Meta-transition matrix of the agent and the MDP transition matrix). The authors in their rebuttal provided some insight into why the assumption is necessary and why it makes sense from a pragmatic point of view. I request author’s to further expand upon the discussion on this assumption in the final copy.

Another concern raised by a critical reviewer focused on novelty, especially in comparison to work in MDPs (mostly due lack of clarity in the definition of active non-stationarity). The critical reviewer, however, was satisfied about other questions and concerns he had raised.

Overall, this is an interesting work with good theoretical and empirical insights and the reviewer pool leans towards acceptance.


**Award:**

No

---

### Decision · Program_Chairs · 2022-09-14

Accept